# A new optical-based technique for real-time measurements of mineral dust concentration in PM10 using a virtual impactor

Authors: Luka Drinovec[1], Jean Sciare[2], Iasonas Stavroulas[2], Spiros Bezantakos[2], Michael Pikridas[2], Florin Unga[2],
Chrysanthos Savvides[3], Bojana Višić[1,4], Maja Remškar[1], Griša Močnik [1,5]

[1]Jozef Stefan Institute, Ljubljana, Slovenia
[2]Climate and Atmosphere Research Center, The Cyprus Institute, Nicosia, Cyprus
[3]Ministry of Labour, Welfare and Social Insurance, Department of Labour Inspection, Nicosia, Cyprus
[4]Institute of Physics Belgrade, University of Belgrade, Belgrade, Serbia
[5]Center for Atmospheric Research, University of Nova Gorica, Ajdovščina, Slovenia

Keywords: mineral dust, aerosol absorption, virtual impactor, PM10

*Correspondence to:* luka.drinovec@ijs.si

**Abstract.** Atmospheric mineral dust influences Earth's radiative budget, cloud formation and lifetime, has adverse health effects, and affects the air quality through the increase of regulatory $PM_{10}$ concentrations, making strategic its real-time quantification in the atmosphere. Only few near-real-time techniques can discriminate dust aerosol in $PM_{10}$ samples and they are based on the dust chemical composition. The on-line determination of mineral dust using aerosol absorption photometers offers an interesting and competitive alternative, but remains a difficult task to achieve. This is particularly challenging when dust is mixed with black carbon, which features a much higher mass absorption cross-section. We build on previous work using filter photometers and present here for the first time a highly time resolved on-line technique for quantification of mineral dust concentration by coupling a high flow virtual impactor (VI) sampler that concentrates coarse particles with an aerosol absorption photometer (Aethalometer, model AE33). The absorption of concentrated dust particles is obtained by subtracting the absorption of the submicron ($PM_1$) aerosol fraction from the absorption of the virtual impactor sample (VI-PM1 method). This real-time method for detecting desert dust was tested in the field for a period of two months (April and May 2016) at a regional background site of Cyprus, in the Eastern Mediterranean. Several intense desert mineral dust events were observed during the field campaign with dust concentration in $PM_{10}$ up to 45 µg m$^{-3}$. Mineral dust was present most of the time during the campaign with an average $PM_{10}$ of about 8 µg m$^{-3}$. Mineral dust absorption was most prominent at short wavelengths, yielding an average mass absorption cross-section (*MAC*) of 0.24 $\pm$ 0.01 m$^2$ g$^{-1}$ at 370 nm and an absorption Ångström exponent of 1.41 $\pm$ 0.29. This *MAC* value can be used as site specific parameter for on-line determination of mineral dust concentration. The uncertainty of the proposed method is discussed by comparing and validating it with different methods.

## 1. Introduction

Atmospheric dust often dominates $PM_{10}$ aerosol mass concentrations in many regions of the world, and is the second most abundant aerosol source at a global scale just after sea spray. Its lifetime in the atmosphere is similar to carbonaceous aerosols (Boucher et al., 2013). Dust particles modify the Earth's radiation balance as they absorb and scatter light, affecting regional climate and precipitation regimes. The net radiative effect of atmospheric dust depends on the interplay between the heating of the atmosphere, due to the increased absorption of sunlight, and cooling due to scattering of sunlight back into space leading to a direct radiative forcing of dust estimated around $-0.1 \pm 0.2$ W m$^{-2}$ (Myhre et al., 2013). Dust deposits on snow and ice increase the ion content in snow and snow water (Greilinger et al., 2018) and they exert a warming influence after deposition (Di Mauro et al., 2015). Desert dust impacts our health and economy. Saharan dust events have been shown to increase morbidity and have negative influence on health mainly through respiratory and cardiovascular effects (Middleton et al., 2008; Perez et al., 2012). The health effects of mineral dust are being considered in the context of regulation (WHO, 2018). Dust soiling of photovoltaics is a significant factor in energy production and decreases their output by up to several percent (Mani and Pillai, 2010). Desert dust is a hazard for air and road transport, can cause electric fields detrimental for communication, and impacts water quality and plants, when deposited, resulting in great economic cost (Middleton, 2017), leading to the fictionalization due to its importance (Herbert, 1965).

Dust particles are often transported from the Sahara over the Mediterranean and southern Europe and can contribute significantly to mass concentration of particles smaller than 10 μm in diameter – $PM_{10}$ (Rodriguez et al., 2001; Vrekoussis et al., 2005). Mineral dust is considered as natural aerosol within the European Air Quality Directive (2008/50/EC) and, as such, can be subtracted from the daily (24h) $PM_{10}$ reported by EU member states, potentially reducing the number of days with $PM_{10}$ exceedances (European Commission, 2011).

Daily time resolution of the described method has been validated with the chemical composition and positive matrix factorization (PMF): the $PM_{10}$ concentration above the daily regional background monthly 40[th] percentile has been shown to correlate well with aluminum (as a tracer of mineral dust), and the mineral dust factor from a PMF analysis (Viana et al., 2010). Methods with higher temporal resolution have the potential to bring considerably more detail and information to the analysis of dust in $PM_{10}$. These methods capture the temporal variability dependent on the synoptic conditions more accurately; they allow the discrimination of long-range transported dust from locally resuspended one (by traffic as an important example); they provide considerably more detail to constrain chemistry-transport models; and can be used in real-time to inform the public and stakeholders and therefore improve adaptation measures. The higher time resolution requires use of novel and innovative approaches and instrumentation.

There are several sampling devices, which allow hourly or sub-hourly sampling of ambient dust aerosols, such as the Streaker sampler, the DRUM (Davis Rotating-drum Unit for Monitoring) sampler (Bukowiecki et al., 2005; Visser et al., 2015) and the SEAS (Semi-continuous Elements in Aerosol Sampler) (Chen et al., 2016). Mass loadings of trace metals collected by these samplers can be analyzed with highy sensitive accelerator-based analytical techniques. However, a major drawback of these analyses is that they require a large commitment of analytical resources and time. Recent technical developments have resulted in more precise, accurate and frequent measurement of ambient metal species, such as the XactTM 625 automated multi-metals analyzer (Fang et al., 2015; Jeong et al., 2016; Phillips-Smith et al., 2017; Cooper et al., 2010).

Dust scatters and absorbs light and its optical properties have been used in on-line measurements to derive the wavelength dependence of the single scattering albedo (*SSA*) (parameterized with the Ångström exponent) as the criterion to characterize Saharan dust events in the high Alpine region (Collaud Coen et al., 2004). The impact of Saharan dust events, showing increased absorption and scattering, was determined in the East Mediterranean (Vrekoussis et al., 2005) and the West Mediterranean (Pandolfi et al., 2011; Pandolfi et al., 2014; Ealo et al., 2016). These measurements with high time resolution have shown that the optical properties can be used to identify dust events. Additionally, combining back-trajectory analysis and the *SSA* wavelength dependence, one can possibly detect local resuspension of dust, which impacts local air quality. However, these methods cannot determine the contribution of desert dust to $PM_{10}$ concentrations in a quantitative manner.

Few studies have reported the potential of using dust aerosol absorption properties to infer their ambient concentrations in $PM_{10}$. These efforts started by using Aethalometers to determine the absorption coefficient attributed to iron compounds in dust, the determination of their mass absorption cross section, and the determination of black carbon and dust in the marine boundary layer (Fialho et al., 2005; Fialho et al., 2006; Fialho et al., 2014). The absorption of dust was due to iron compounds which were quantified using instrumental neutron activation analysis. Zhang et al. (2008) used thermal-optical reflection to measure the carbonaceous fraction and proton induced X-ray emission for elemental analysis, and again used iron as the dust tracer to separate the contributions of these two light-absorbing aerosol components. These publications systematically biased the absorption coefficients too high due to the assumption that the attenuation of light in the filter is due to non-filter-enhanced absorption, neglecting the enhancement due to the scattering in the filter matrix. Using different influence of iron-containing mineral dust and black carbon on $SSA$ at different wavelengths and contrasting fine and coarse fractions, Derimian et al. (2008) quantified the iron concentrations in mineral dust. Lately, more sophisticated techniques using filter photometers were employed to determine the mineral dust absorption coefficients, mass absorption cross-sections and dust $SSA$. Caponi et al. (2017) used the multi-wavelength absorbance analyzer to determine the absorption coefficients at multiple wavelengths and obtain the absorption Ångström exponents and mass absorption cross-sections in a chamber study. The chamber study was also used to determine the filter enhancement in Aethalometers challenged with dust (Di Biagio et al., 2017) and then use these parameters to determine the optical properties as a function of iron content for different dust samples from all over the world (Di Biagio et al., 2019).

Additionally, quantitative determination of ambient concentration of mineral dust has been performed in the mixture of Saharan dust and carbonaceous matter in a wildfire plume (Schauer et al., 2016). These two may be internally mixed (Hand et al., 2010). The relationship between the columnar optical properties and the in-situ ones during dusty and dust-free days due to the mixing of the dust with the dominant local air pollution is challenging to interpret (Valenzuela et al., 2015).

Previous work has used two-component models to infer dust concentrations sampling ambient air on a filter in filter absorption photometers. However, the determination of the optical absorption of pure mineral dust - when being externally mixed with black carbon - is more difficult because black carbon features a much higher mass absorption cross-section, obscuring the smaller contribution of dust to absorption. Enriching the aerosol coarse fraction, and hence increasing the contribution of weakly absorbing dust, may represent an innovative alternative way to increase dust aerosol absorption relative to black carbon.

We present here an improvement in real-time detection of mineral dust in ambient $PM$ by concentrating the coarse particle fraction with a high-volume virtual impactor system similar to the one reported by Sioutas et al. (1994), and coupled with an aerosol absorption monitor (Aethalometer model AE33). We demonstrate its performance at a regional background site, frequently impacted by Saharan dust.

First the enhanced absorption of coarse particles is determined from the difference of absorption measured by Aethalometers with the virtual impactor and $PM_1$ inlet, respectively. This parameter is divided by the enhancement factor calculated from the particle size distributions, yielding the absorption of coarse particles. The correlation between mineral dust absorption and reference mineral dust concentration provides us with the mass absorption cross-section of the mineral dust, which is then used to determine mineral dust concentration with high time resolution.

## 2. Materials and methods

### 2.1. Field campaign site description

Field validation took place at the Cyprus Atmospheric Observatory (CAO) between 1 April 2016 and 31 May 2016. This field campaign was organized as part of the European projects ACTRIS (Aerosols, Clouds and Trace Gases Research Infrastructure) and BACCHUS (Impact of Biogenic versus Anthropogenic emissions on Clouds and Climate; towards a Holistic UnderStanding). CAO is situated at a regional background site on the foothills of mount Troodos (35.04N; 33.06E; 535 m a.s.l) in the centre of Cyprus, an island located in the Eastern Mediterranean - Levantine basin. Lying in close proximity to the Middle-East/North Africa (MENA) region, Cyprus is often influenced by air masses carrying mineral dust particles, originating from either the Saharan Desert or the Middle East (Mamouri et al., 2013; Pikridas et al., 2018). During this field campaign, a large suite

of in-situ and remote sensing instrumentation was deployed at ground level and onboard Unmanned Aerial Vehicles in order to better characterize the influence of desert dust on Ice Nuclei (Schrod et al., 2017; Marinou et al., 2019), LIDAR retrieval of vertically-resolved *PM* (Mamali et al., 2018), and performance of miniaturized light absorption sensors (Pikridas et al., 2019). More information on the climatology of air masses origin and PM at the Cyprus Atmospheric Observatory can be found in Pikridas et al., 2018).

## 2.2. Instrumentation

Real-time aerosol absorption of the dust-containing coarse fraction was determined by subtracting aerosol absorption of black carbon-containing submicron aerosols from the absorption of the concentrated coarse aerosols from the outlet of a virtual impactor (VI). A detailed description of this instrumental set-up as well as
complementary aerosol instrument is provided in the following.

### 2.2.1. Virtual Impactor design

Based on a design similar to the one reported by Sioutas et al. (1994), our virtual impactor is sampling ambient
air (TSP) at a total flow rate of 100 l min$^{-1}$. The major flow ($F_{in}$ = 98 l min$^{-1}$) is carried out of the VI by a large capacity pump while the coarse particles are inertially impacted (enriched) into the minor flow ($F_{out}$=2 l min$^{-1}$) of the VI connected to the absorption photometer. The enrichment of coarse particles in the minor flow is a function of the ratio of the two (major/minor) flows; its efficiency depends also on the design and manufacturing of the VI. For that reason, the VI was thoroughly tested and characterized in the lab in order to
estimate the concentration efficiency (*CE*) of coarse particles as a function of aerodynamic particle size (using NIST polystyrene-latex (PSL) spheres with nominal aerodynamic sizes from 0.7 to 10 μm) and two different flow rate ratios (19 and 50, respectively). The laboratory characterisation of the VI is described in the Supplement S1 while the methodology used to reconstruct the size distribution of concentrated coarse particles (in the minor flow $F_{out}$) is presented in section 3.1.

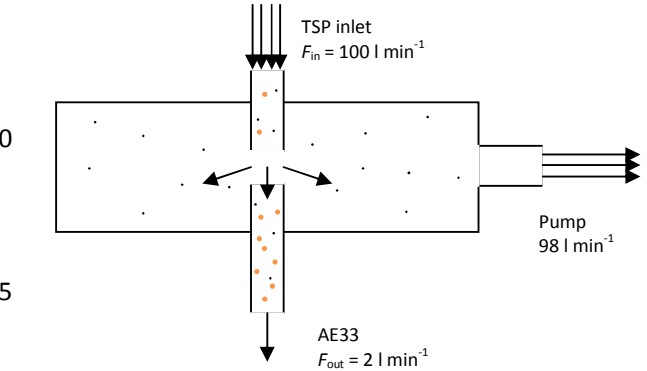

TSP inlet
$F_{in}$ = 100 l min$^{-1}$

Pump
98 l min$^{-1}$

AE33
$F_{out}$ = 2 l min$^{-1}$

**Figure 1. Principle of the Virtual Impactor (VI) operation.**


### 2.2.2. Aerosol absorption and light scattering coefficients

Three Aethalometers model AE33 (Magee Scientific, USA) were used during the field campaign with different inlet setups: one sampling through a PM1 sharp-cut cyclone (BGI Inc., Model SCC 1.197), one sampling through a custom-made total suspended particles (TSP) inlet and a third one sampling through the VI described above
(Section 2.2.1). The first two instruments (with PM1 and TSP) were sampling at a 5 l min$^{-1}$ flow rate, while the third was sampling through the VI at a 2 l min$^{-1}$ flow rate. This flow rate of 2 l min$^{-1}$ was selected so as to increase concentration efficiency (CE) of the VI and consequently increase the absorption signal of dust aerosols.

The Aethalometer AE33 measures attenuation of light by two samples collected at different flow rates. This results in two sample spots that feature different attenuation values. The measurement of light transport through the sample-laden filter is non-linear and the measurements using two sample spots allow the on-line compensation of the nonlinearity of the black carbon measurement (Drinovec et al., 2015). Given that the on-line filter loading compensation was not working efficiently for the AE33 coupled with the virtual impactor (see

section 3.3, below), the obtained data was compensated using fixed filter loading compensation parameter *k* values as described in the Supplement S2.

Absorption coefficient ($b_{abs}$) was calculated from the attenuation coefficient ($b_{atn}$) using the value of the multiple scattering parameter *C* of 2.57:


$$b_{\text{abs}} = \frac{b_{\text{atn}}}{C} \qquad\qquad (1)$$

The calculation of the absorption coefficient was updated from the Drinovec et al. (2015) following the WMO guideline (WMO, 2016): we updated the value of the filter multiple-scattering parameter *C*. The multiple-
scattering parameter *C* in Drinovec et al. (2015) determined the AE33 filter *C* values relative to the value of the quartz filter, used in older AE31 instruments. This AE31 value was assumed to be 2.14 (Weingartner et al., 2003), but it was later recommended to use an AE31 value of 3.5 (WMO, 2016). The parameter *C* = 1.57 used for the AE33 filter (Drinovec et al., 2015) was renormalized using the same factor resulting in a new value *C* = 2.57. The mass absorption cross-section $\sigma_{air}$ for black carbon was adjusted in the inverse manner to obtain the
same *BC*. The new mass absorption cross section for black carbon $\sigma_{air}$ at 880 nm is now 4.74 $m^2\ g^{-1}$ instead of 7.77 $m^2\ g^{-1}$:

$$BC = \frac{b_{\text{abs}}}{\sigma_{\text{air}}} = \frac{b_{atn}}{C \cdot \sigma_{\text{air}}}. \qquad\qquad (2)$$

The Aethalometers were intercompared in the laboratory before the campaign. The correlation slope for 1 minute resolution data differed less than 4% between the instruments with $R^2$ = 0.996. The analysis of the actual uncertainty of Aethalometer measurements during the campaign is presented in the Supplement S3.

Total scattering and back-scattering coefficients ($b_{scat}$ & $b_{bscat}$) of the ambient (TSP) aerosol were monitored
continuously using a three-wavelength (450, 550 and 700 nm) integrated nephelometer (TSI Inc., model 3563; Anderson and Ogren, 1998). The nephelometer was sampling through a vertical, straight sampling line, coupled with a TSP inlet, a Nafion dryer, and measuring at 1-minute time resolution. Calibration was conducted using $CO_2$ as a high, and zero air as a low span gas prior to the field campaign. This nephelometer went through a successful inter-comparison exercise at the European Center for Aerosol Calibration (ECAC-report-IN-2015-1-5,
2016) ahead of the instrument's field deployment. Nevertheless, due to miscalibration of the green channel, 550 nm measurements were excluded from the analysis. Single scattering albedo (SSA) was calculated at 450 and 700 nm using the total scattering coefficient from nephelometer and absorption coefficient obtained from AE33 by linear interpolation of absorption coefficients from adjacent wavelength pairs. Single scattering albedo Ångström exponent (*SSAAE*) was calculated from $SSA_{450\ nm}$ and $SSA_{700\ nm}$.


### 2.2.3. Other in-situ aerosol instrumentation

Dried particle number size distributions (PNSDs) were measured using a TSI Inc. Aerodynamic Particle Sizer (APS, model 3321). The APS measures PNSDs in the 0.5 – 20 μm aerodynamic diameter size range at a 5-minute
temporal resolution. The APS was sampling at a total flow rate of 5 l $min^{-1}$ through a straight vertical sampling line, a Nafion dryer, and a TSP inlet, identical to the nephelometer. Aerosol mass concentration for fine ($PM_{2.5}$) and coarse aerosols ($PM_{10-2.5}$) was measured using a Continuous Dichotomous Ambient Particulate Monitor (Thermo Scientific, 1405-DF TEOM-FDMS system) deployed at the Agia Marina Xyliatou EMEP station, collocated with CAO, at a 1-hour temporal resolution (see more at Pikridas et al., 2019).


### 2.3. Filter sampling and analysis

### 2.3.1 Filter sampling

Aerosol samples were collected during the field campaign at a flow rate of 2.3 $m^3\ h^{-1}$ on pre-weighed filters (Pall Tissuquartz 2500 QAT-UP) using two autonomous filter samplers (Leckel SEQ 47/50) for determination of
mass concentration ($PM_{2.5}$ and $PM_{10}$, respectively) with 24-h time resolution from midnight to midnight according to local standard time. Particle mass concentration ($PM_{2.5}$ and $PM_{10}$) on the filter substrates was determined gravimetrically before and after the sampling, under constant conditions dictated by protocol

EN12341 with the use of a 6 digits precision analytical balance (Mettler Toledo, Model XP26C). According to that protocol filters were subjected to 45-50% relative humidity at 20 $\pm$ 1$^o$ C for 48 hours.


### 2.3.2 Aerosol chemical mass closure

Filter samples were subsequently analyzed for major ions by ion chromatography (Thermo, Model ICS5000) following the protocol reported in Sciare et al. (2011) and complying with the European committee for standardization for the measurement of anions and cations in PM2.5 (EN 16913:2017) and elemental carbon

concentration (*EC*) and organic carbon concentration (*OC*) with a Sunset Lab Instrument, the EUSAAR_2 thermo-optical protocol (Cavalli et al., 2010) and complying with the European committee for standardization (EN 16909:2017). Quality of ion measurements is checked at the bi-annual intercomparison studies performed in the framework of the EMEP and WMO networks while quality of EC/OC measurements is confirmed on annual basis in the framework of ACTRIS network.


Estimation of dust in $PM_{2.5}$ and $PM_{10}$ was performed following the methodology proposed by Sciare et al. (2005) for a regional background (Crete Island) located in the Eastern Mediterranean assuming a constant Calcium-to-dust ratio of 0.12. Reconstruction of *PM* from chemical analyses versus *PM* from gravimetry is reported in Supplement S6 and shows very good correlation ($R^2$ = 0.99) and slope close to one, supporting the

consistency and robustness of our calculation of mineral dust in PM.

### 2.3.3 Trace metal analysis using inductively coupled plasma mass spectrometry (ICP-MS)

An acid microwave digestion procedure was applied to the $PM_{10}$ filters followed by inductively coupled plasma mass spectrometry (ICP-MS, Thermo Electron X Series) to measure metal concentrations of Al, V, Cr, Mn, Fe, Ni,

Cu, Zn, Cd, and Pb, following the procedure from Poulakis et al. (2015).

### 2.3.4 Scanning electron microscopy - energy dispersive X-ray (SEM-EDX) analysis

Scanning electron microscopy measurements were performed at Jožef Stefan Institute using SEM model Supra

35 VP (Carl Zeiss, Germany). Measurements were performed on punches of PM10 filters that were attached to the sample holder through a double-sided carbon tape. The filters were previously sputter-coated with a thin gold film (with Au nanoparticle approximate size of 10 nm) using an SCD 005 cool sputter coater (BAL-TEC GmbH, Leica Microsystems, Wetzlar, Germany). The microscope was equipped with the energy dispersive spectroscopy module (EDX, Oxford INCA 400, Oxford Instruments Analytical, UK), which was operated at an

accelerating voltage of 20 kV.

### 2.4. Data coverage

The measurement campaign took place in April and May 2016. Due to technical reasons not all of the instruments were running throughout the campaign (Table 1). This limited some of the analysis to a shorter

time periods with most of the data available between 14 April and 6 May.

**Table 1. Data coverage for on-line instrumentation and filter sampling.**

| Instrument | Available data |
| --- | --- |
| AE33 | 4 Apr 2016 – 31 May2016 |
| VI | 14 Apr 2016 – 6 May 2016 |
| TEOM-FDMS | 1 Apr 2016 – 31 May 2016 |
| Nephelometer | 14 Apr 2016 – 31 May 2016 |
| APS | 1 Apr 2016 – 30 Apr 2016 |
| 24h filter samples | 1 Apr 2016 – 31 May 2016 |


## 3. Results and discussion

The absorption of the fine aerosol fraction, dominated by black carbon, is usually much larger compared to the absorption in the coarse fraction, which contains mineral dust. In other words, $b_{abs,TSP}$ is expected to be close to $b_{abs,PM1}$. Subtracting these two signals close in absolute values (but with large uncertainties), would result in a close-to-zero number associated with a large measurement error. The proposed VI method takes advantage of the concentration of coarse particles using a virtual impactor to enhance the coarse fraction in the sample, and subtracts the absorption of the fine fraction (as the VI sample contains the same amount of fine fraction as an ambient sample). To calculate the absorption of the coarse fraction, the concentration efficiency of the virtual impactor must be taken into account.

The methodology to derive real-time concentration of dust in $PM_{10}$ is presented as per the follow: the robustness of the system (VI+AE33) is tested first in the field for a period of 1 month in Section 3.1. The enhancement factor (EF) downstream of the virtual impactor is calculated in Section 3.2. Real-time absorption of ambient dust aerosols is calculated in section 3.3. It is corrected for 1) the influence of black carbon measured by a co-located Aethalometer AE33 equipped with $PM_1$ inlet, and 2) the enhancement factor of the VI. Real-time dust concentration of $PM_{10}$ (Section 3.4) is then derived by dividing the absorption of dust aerosols calculated in section 3.3 with a mass absorption cross-section (*MAC*) for dust calculated using filter-based chemical analyses.

### 3.1. Field campaign overview

Optical, physical, and size-resolved chemical properties of ambient aerosols at the Cyprus Atmospheric Observatory were characterised continuously in April and May 2016 using several online and offline methods as illustrated in Figure 2. During the campaign a total of four intense events (16 and 26 April, 1 and 15 May 2016) were detected with increased *PM* concentration of the coarse fraction concurrent with increase of the light scattering coefficient, but no correlation with black carbon concentration (Figure 2). By combining light absorption and scattering measurements it is possible to identify the mineral dust events as the periods during which the single scattering albedo Ångström exponent (*SSAAE*) becomes negative, indicating the presence of mineral dust (Collaud Cohen et al., 2004).

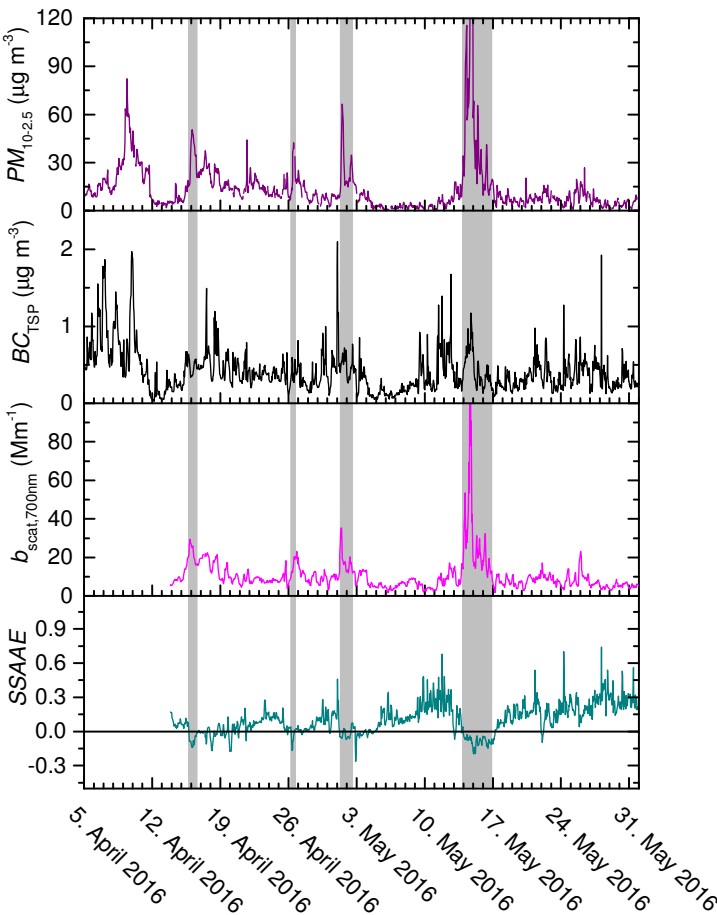

**Figure 2. Time series of the mass concentration of coarse particles (*PM*<sub>10 -2.5</sub>) obtained by TEOM-FDMS, black carbon concentration (*BC*) for total suspended particles (TSP) obtained by Aethalometer AE33, the light scattering coefficient at 700 nm obtained by nephelometer and single scattering albedo Ångström exponent (*SSAAE*). The four periods with negative *SSAAE* during the campaign are shadowed.**

Figure 3 reports continuous measurements of aerosol absorption during the field campaign for three different aerosol size fractions that were achieved using different inlets for three Aethalometers AE33 running in parallel: 1) a $PM_1$; 2) a total suspended particle (TSP), and 3) a virtual impactor (VI). During the first days of the campaign, the VI was operated manually for several periods of few hours in order to perform several tests related to the collection and light absorption detection of the coarse faction. The VI was set ON continuously from 14[th] April till 6[th] May. Given the strong sensitivity of dust aerosol absorption in the UV range compared to black carbon (see Section 3.3), the 370 nm channel was selected here to compare aerosol absorption measurements from the three Aethalometers.

As shown in Figure 3 the absorption in TSP is closely related to the one of $PM_1$. The differences are inside the measurement uncertainty of the Aethalometers (Supplement S3).  On contrary, the absorption measured when the virtual impactor was ON shows very high values and very poor correlation with the other absorption measurements. During the periods when the VI pump was not operating, the aerosol absorption agrees well with the one using the TSP showing that the enhancement of aerosol absorption is entirely related to the enhancement of the coarse fraction.

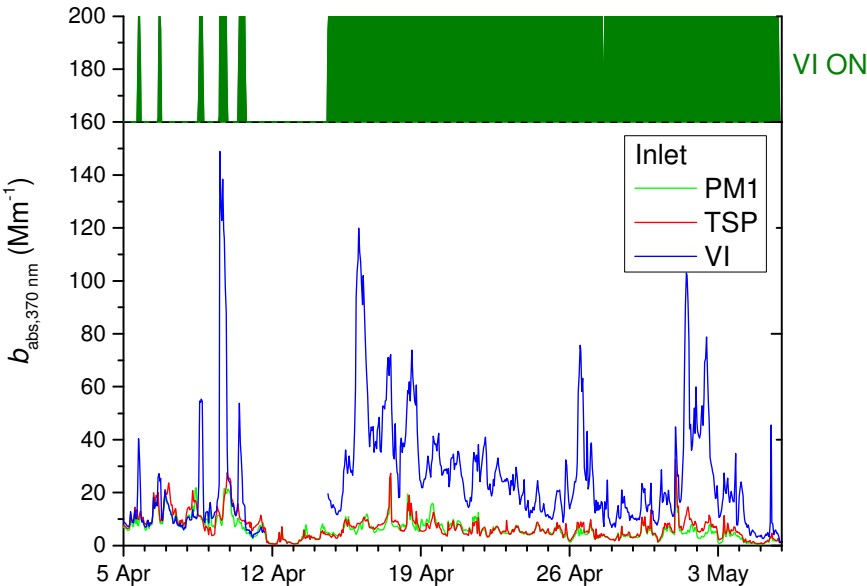

**Figure 3. Time series of the aerosol absorption coefficient at 370 nm for three Aethalometers AE33 running in parallel and equipped with different inlets: PM$_1$, total suspended particle (TSP) and virtual impactor (VI). The periods when VI was operating are marked in green (VI=ON).**

### 3.2. Experimental characterization of the enhancement factor of coarse particles using the virtual impactor

The concentration efficiency of the virtual impactor depends on the aerodynamic particle diameter. For that reason an Aerodynamic Particle Sizer (APS, model 3321) was used both for analysis of aerodynamic particle size distribution during the campaign and virtual impactor characterization (Supplement S1). The maximum concentration efficiency (CE) was obtained for $F_{in}/F_{out}$ ratio of 50 (Table S1). Using the minimum sample flow of AE33 of 2 lpm and maximum flow of the virtual impactor pump we obtained the maximum $F_{in}/F_{out}$ ratio of 50. As shown in Table S1 for $F_{in}/F_{out}$ ratio of 50, the concentration efficiency of aerosols with aerodynamic diameters of 1 μm and below is close to unity. This result is expected for a virtual impactor which principle is based on the concentration of large aerosols. Consequently, the black carbon fraction - mainly located in the submicron mode - is not enriched in the VI. On contrary, dust aerosols - mostly located in the coarse aerosol fraction (above 1 μm diameter) are concentrated efficiently by the virtual impactor (Figure 4).

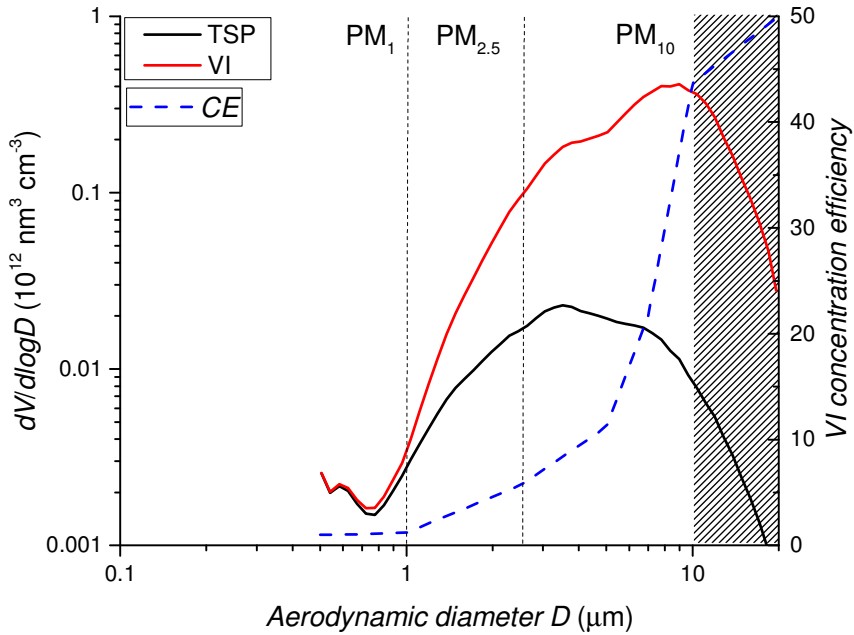

**Figure 4. Average aerodynamic volume size distribution spectrum measured by APS between 1 April 2016 – 30 April 2016 (black line) and the predicted spectrum of aerosol concentrated using the virtual impactor (red line). Blue line represents the virtual impactor concentration efficiency measured in laboratory (Supplement S1). Shaded area denotes the particles larger than 10 $\mu$ m.**

The enhancement factor (*EF*) of the VI is defined as a multiplication factor that reflects the enrichment of the coarse fraction downstream of the VI. EF was determined experimentally during the field test using the Aerodynamic Particle Sizer (APS) to derive both the volume concentration of the unperturbed sample (*V*) and volume concentration enhanced using a virtual impactor ($V_{VI}$):

$$V = \int \left( \frac{dV}{dlogD} \right) * dlogD \qquad , \qquad (3)$$

$$V_{VI} = \int \left( \frac{dV}{dlogD} \right) * CE * dlogD \qquad , \qquad (4)$$

 where *D* is the particle aerodynamic diameter and *CE* the concentration efficiency of the VI as characterized in supplement S1. The enhancement factor is then calculated as:

$$EF = \frac{V_{VI}}{V} \qquad . \qquad (5)$$

 The average aerosol volume concentration size spectrum obtained during the campaign by the APS is presented in Figure 4 (black line), along with the spectrum calculated for the virtual impactor (red line), using the concentration efficiency determined during the laboratory campaign (Table S1). For ambient aerosol we observe a mode around 3.5 µm. Because the virtual impactor is more efficient towards larger particles, the ambient volume size distribution is not reproduced downstream of the VI, which shows a maximum around 9  µm. The collection efficiency of particles larger than 10 µm in the Aethalometer AE33 downstream the VI is expected to be rather low due to losses in tubing and sample lines inside the Aethalometer. Overall, the uncertainty associated with the enhancement factor remains difficult to assess in the aerosol range close to 10 µm diameter which particles are usually difficult to collect in a quantitative way.

The enhancement factor of the VI defined in Eq. 5 depends on the ambient aerosol volume size distribution measured by the APS, which changes over time. Figure 5 reports the temporal variation of *EF* during the field campaign at 5 min time resolution. There are time intervals with stable *EF* of approximately 9 (11 April 2016 – 13 April 2016), also we can observe some peaks with *EF* as high as 16. For the $PM_{10}$-$PM_1$ particles we obtained a campaign average VI enhancement factor:

*EF* = 11 $\pm$ 2.

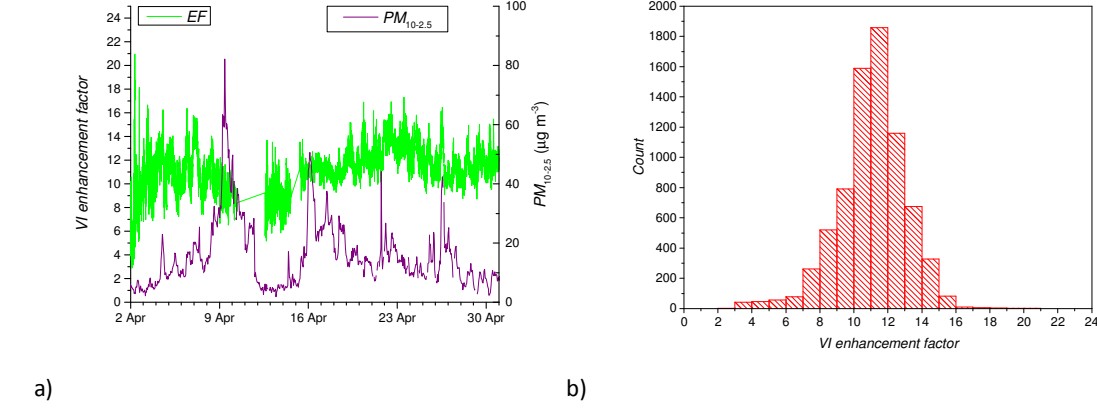

a)                                                                                                 b)

**Figure 5. Time series of the virtual impactor enhancement factor calculated from aerodynamic size distributions measured by APS (a) for the PM$_{10}$ aerosol fraction. The enhancement factor frequency distribution is shown on the right (b).**

### 3.3. Calculation of the absorption coefficient of coarse particles

The principle of the virtual impactor operation allows for concentration of the coarse particles while the fine particles remain present at the same amount as in the ambient air. To determine the absorption induced by ambient mineral dust, we need to subtract the fine particle absorption signal (dominated by black carbon) from the total virtual impactor absorption and normalize by the enhancement factor, following the equation:

$$b_{abs,PM10-1} = \frac{b_{abs,VI} - b_{abs,PM1}}{EF} \qquad , \qquad (6)$$

where $b_{abs,PM10-1}$, $b_{abs,VI}$ and $b_{abs,PM1}$ represent absorption coefficients of dust in ambient conditions, aerosols downstream of the virtual impactor and submicron aerosols, respectively. Because absorption of PM$_1$ fraction is dominated by black carbon, it is essential to compensate absorption data for the filter loading effect (Drinovec et al., 2017). If the compensation parameter is wrong by 0.005 this can result in over- or under-estimation of $b_{abs}$ by up to 60% at 370 nm and by 25% at 950 nm. For the Aethalometer with the PM$_1$ inlet, the absorption data is sufficiently compensated by the built-in dual-spot algorithm. For the Aethalometer connected to the virtual impactor, the method was not able to measure accurately the loading effect due to the presence of coarse particles (Supplement S2). The main reason for this behaviour lies in the fact that a single particle (deposited on one of the two spots) potentially causes significant absorption only in one of the two measurement spots. This requires an application of off-line compensation using fixed values of the compensation parameters (Supplement S2).

The absorption induced by dust (Eq. 6) during the field campaign was calculated for each of the 7 wavelengths of the Aethalometer AE33 and averaged, as shown in Figure 6. The spectral dependence of absorption by mineral dust shows an increase at shorter wavelengths, significantly deviating from the Ångström exponent of 1. The best discrimination between the mineral dust particles and black carbon is achieved at the lowest wavelength 370 nm, which is the wavelength that has been selected in our procedure to derive the absorption and the atmospheric concentration of mineral dust.

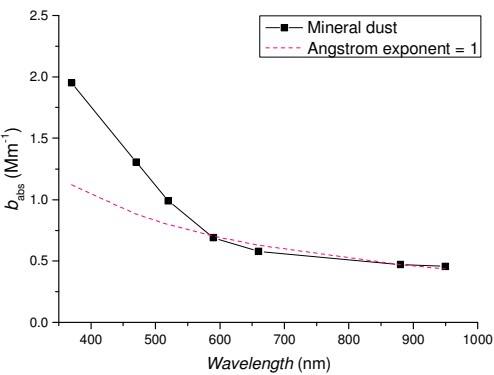

**Figure 6. Averaged absorption coefficient during the campaign for ambient dust (as calculated in Eq. 6) in the coarse**
**fraction of aerosols as calculated from the difference of the absorption coefficients measured with a virtual impactor and**
**a PM₁ inlet, and divided by the average enhancement factor (black line). The dotted red line shows a theoretical curve**
**with Ångström exponent of 1, extrapolated from measurements at 880 nm.**

Campaign averages show the mean value for absorption at 370 nm being higher for TSP compared to PM$_1$ inlet
(Table 2). The absorption signal reported in this table for the AE33 behind the VI is more than a factor of 4
higher compared to AE33 with the PM$_1$ inlet and is due to the concentrated mineral dust in the coarse fraction.
The average absorption coefficient of ambient dust as calculated using Eq. 6 was 2.0 $\pm$ 2.1 Mm$^{-1}$. Similarly to
absorption, *AAE* shows higher value for TSP compared to PM$_1$ inlet, as the mineral dust in the coarse fraction
increases absorption in UV & blue part of the spectrum. As expected there is high variability both for
absorption and *AAE* during the campaign. The average difference between absorption for TSP and PM1 inlets is
lower than expected from the (VI-PM1)/2 absorption value. This is a consequence of high measurement
uncertainty of up to 18% during the campaign (as estimated comparing Aethalometers with different inlets,
Supplement S3). Mineral dust absorption calculated from the difference between $b_{abs,370nm}$ for TSP and PM1 of
0.6 Mm$^{-1}$ has an uncertainty 2.1 Mm$^{-1}$.

**Table 2. Average absorption coefficient at 370 nm, the absorption Ångström exponent (*AAE*) and their variation during**
**the campaign (15 April 2016 and 6 May 2016). *AAE*s were calculated as averages of one hour values for the wavelength**
**pair of 370 and 950 nm.**

| Sample | $b_{abs,370nm}$ (Mm$^{-1}$) | *AAE* |
|---|---|---|
| PM1 | 5.6 $\pm$ 3.3 | 1.22 $\pm$ 0.16 |
| TSP | 6.1 $\pm$ 3.4 | 1.30 $\pm$ 0.18 |
| VI | 27.6 $\pm$ 20.1 | 1.38 $\pm$ 0.25 |
| VI-PM1 | 22.0 $\pm$ 23.4 | 1.41 $\pm$ 0.29 |
| (VI-PM1)/11 | 2.0 $\pm$ 2.1 | 1.41 $\pm$ 0.29 |

### 3.4. Determination of mineral dust mass absorption cross-section

For the determination of the mineral dust mass absorption cross-section, we need to establish the mineral dust
concentration in our samples. For this purpose, we performed mass closure on 24-hour $PM_{10}$ and $PM_{2.5}$ filter
samples (see Supplement S6). Mineral dust concentration was determined from calcium concentration,
assuming 12% Ca in mineral dust. Since the virtual impactor concentrates larger particles with higher efficiency
(this is where we expect to have the largest contribution of mineral dust), we used the coarse fraction $PM_{10-2.5}$
for the calibration (Figure 7).

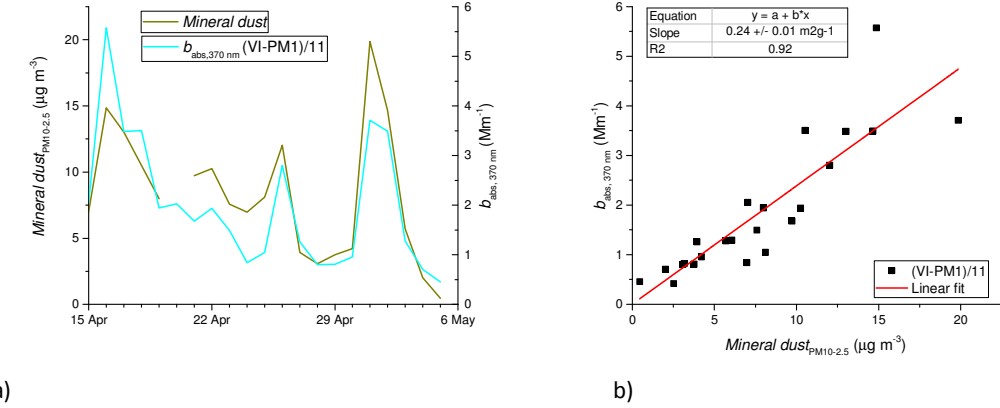

a)                                                                                                          b)

**Figure 7. The time-series (a) and the correlation (b) between the absorption coefficient of coarse particles obtained using
VI-PM1 method versus mineral dust concentration in the coarse fraction.**

The correlation between mineral dust absorption and filter measurements is very good with $R^2$ of 0.92 showing
good agreement between the methods. The mineral dust mass absorption cross-section was obtained from the
regression between mineral dust mass and the coarse fraction absorption coefficient at 370 nm:

$MAC_{\text{mineral dust,370nm}}$ = 0.24 $\pm$ 0.01 m$^2$ g$^{-1}$.

This *MAC* value, obtained with 24-h time resolution, allows us to calculate mineral dust concentrations with the
high time resolution of the absorption measurements:

$$Mineral\ dust_{PM_{10-2.5}} = \frac{(b_{\text{abs,370nm,VI}} - b_{\text{abs,370nm,PM1}})}{EF \cdot MAC_{\text{mineral dust,370 nm}}} \qquad (7)$$

The concentration of coarse particles and its composition show a huge variability during the campaign (Figure
8a): the average mineral dust concentration was 8.1 μg m$^{-3}$ with peaks up to 45 μg m$^{-3}$. On average, mineral
dust represented about one half of the coarse fraction. The average BC was much lower at 0.39 μg m$^{-3}$. Due to
its much higher mass absorption cross-section ($MAC_{\text{BC,370nm}}$ = 11.2 m$^2$ g$^{-1}$; 47 times higher compared to mineral
dust), the absorption of black carbon dominated the aerosol absorption in Cyprus during the campaign and
mineral dust absorption could not be detected directly. Using the virtual impactor allows us to concentrate the
dust and measure its absorption coefficient and determine its mass absorption cross-section with a low
uncertainty.

The Ångström exponent of the fine fraction oscillates between 1 and 1.5 (Figure 8b). The lower values
correspond to *BC* peaks, originating from local traffic and other efficient combustion sources. The higher values
are a mixture of mineral dust and local pollution. The Ångström exponent of the coarse fraction reaches value
of 2.1 during intense mineral dust periods. After these events the *AAE* value drops slowly and reaches a value
of 1.2 during the period with low presence of mineral dust (around 27 April 2019 – 1 May 2019).

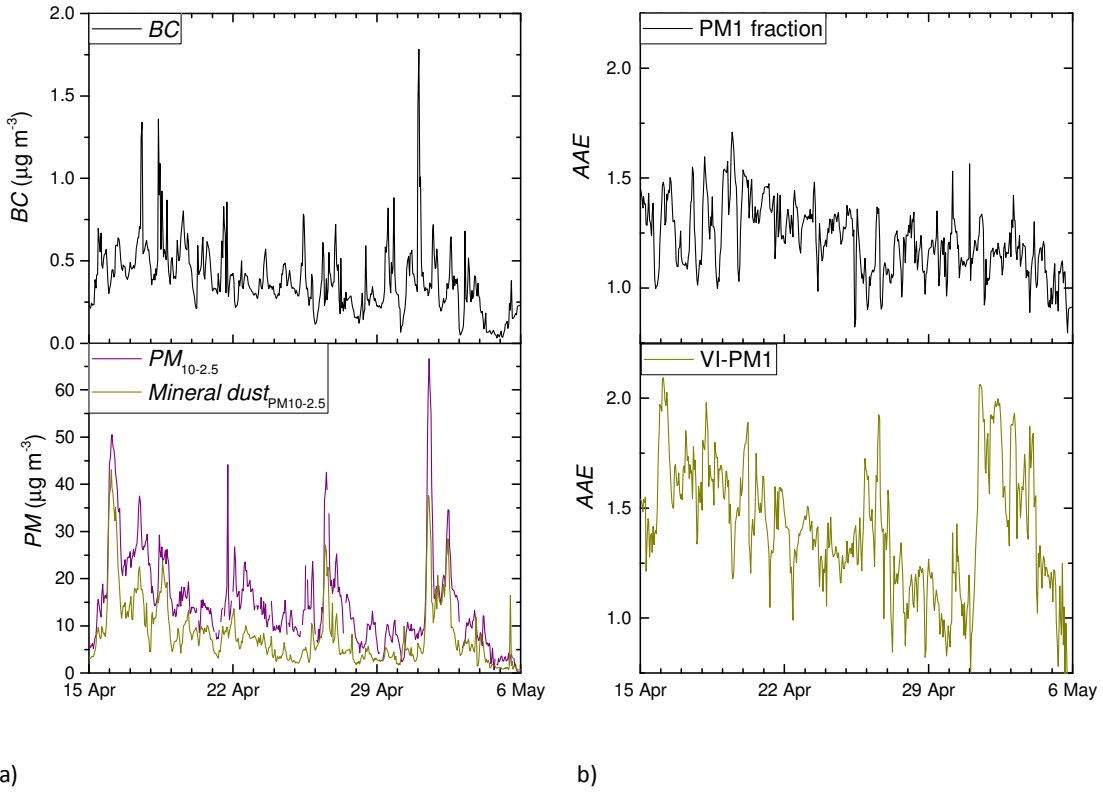

a)                                                                    b)

**Figure 8. Time series of *BC* in fine fraction and mineral dust determined using VI-PM1 method (a) and Ångström exponent for fine and coarse fraction of the aerosol (b).**

The knowledge of the virtual impactor enhancement factor allowed us to calculate the average *MAC* for mineral dust during the Cyprus campaign. Reportedly, its value depends mostly on the absorption of iron oxides (Sokolik and Toon, 1999, Alfaro et al., 2003; Fialho et al., 2005; Fialho et al., 2006; Fialho et al., 2014; Caponi et al., 2017; Di Biagio et al., 2019). During the campaign we obtained 1.9% iron in $PM_{10}$. If we take into account that mineral dust represented about one half of $PM_{10}$ (Supplement S6), we get a good agreement with iron concentrations measured for mineral dust from Middle East of 3.15% ‒ 3.5% (Linke et al., 2006), 3.8% - 5% (Caponi et al., 2017) and Sahara of 3.6% - 6.6% (Caponi at al., 2017). Surprisingly our $MAC_{\text{Mineral dust,370nm}}$ = 0.24 $m^2\,g^{-1}$ is much larger compared to mineral dust from Saudi Arabia of 0.09 $m^2\,g^{-1}$, Libya of 0.089 $m^2\,g^{-1}$ and Algeria of 0.099 $m^2\,g^{-1}$ (Caponi et al., 2017) or North-Eastern Africa of 0.099 $m^2\,g^{-1}$ (Fialho et al., 2006). Higher *MAC* goes along with the low Ångström exponent value of 2.1 obtained for the fresh mineral dust reaching Cyprus. This value is lower than AAE of 2.8. ‒ 4.1 reported for Middle East (2.8 ‒ 4.1) and Sahara (2.5 ‒ 3.2) by Caponi et al. (2017) or 2.9-4 for North-Eastern Africa (Fialho et al., 2005; Fialho et al., 2006). Differences in *MAC* values and the Ångström exponent can be an indicator that the coarse fraction of mineral dust is contaminated with black carbon, with the mixing occurring in or close to the source regions much earlier than mineral dust reached Cyprus. Simulations show that black carbon stuck to the mineral dust particles can severely change its optical properties, but the effect depends on the particle size (Scarnato et al., 2015). Because of the differences in dust mineral composition and contamination with BC we expect MAC to be source region specific.

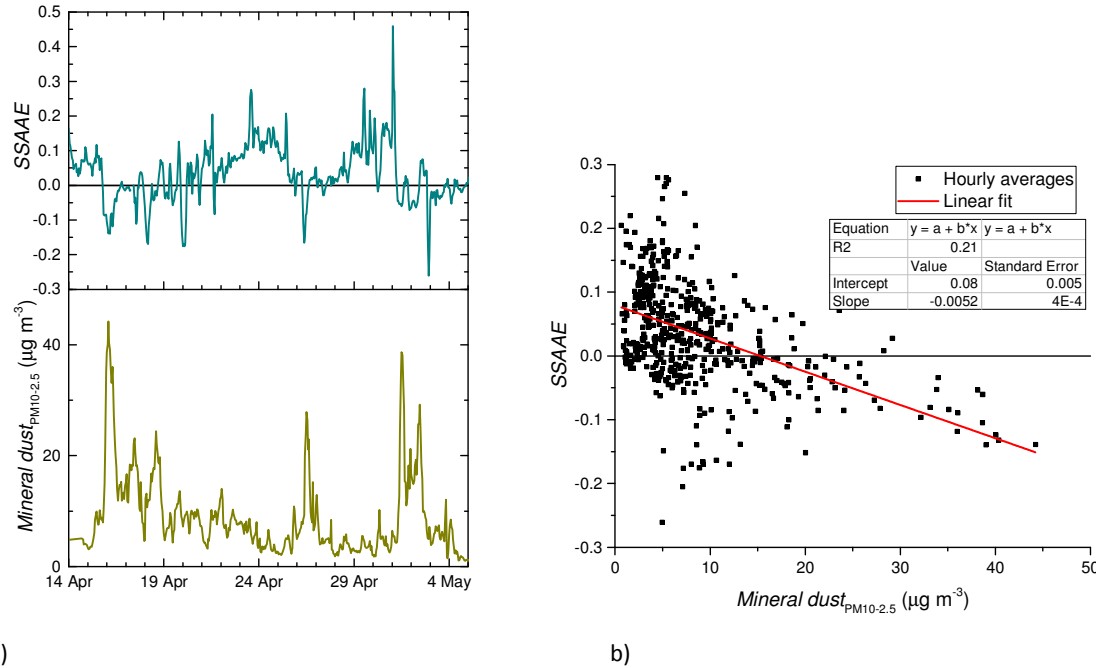


a)                                                                                                           b)

**Figure 9. Time-series (a) and correlation (b) of the single scattering albedo Ångström exponent (*SSAAE*) and mineral dust concentration.**


The determination of the mineral dust concentration was tested using the Collaud Coen et al. (2004) method for the qualitative determination of Saharan dust events. As expected the peaks in mineral dust concentration correspond to the periods featuring negative values of single scattering albedo Ångström exponent. It is shown (Figure 9) that *SSAAE* becomes negative when the mineral concentration becomes larger than 15 µg m$^{-3}$. The

correlation between the mineral dust (determined from the chemical composition, notably calcium ions) and *SSAAE* is not perfect because of the contribution to scattering from other aerosol components (organics, sulphates …) which amount to about one half of the aerosol mass, and the absorption of organics. While the *SSAAE* method provides an identification of dust events, the VI-PM1 method allows for the quantitative determination of mineral dust - even at high black carbon concentrations in the fine fraction.


**3.7. Uncertainty of the VI-PM1 method for the determination of mineral dust concentrations in *PM$_{10}$***

The uncertainty in the determination of mineral dust concentration using VI-PM1 method arises from the measurement uncertainties, variability of optical and chemical properties of mineral dust and potential

systematic biases of the method itself. Since the VI-PM1 method is calibrated using mineral dust in the coarse fraction only, 6% lower values compared to total mineral dust in *PM$_{10}$* are reported. This bias can be avoided by using a correction factor.

The uncertainty of 10% in the determination of the attenuation coefficient at 880 nm by the Aethalometer

AE33 was reported (Drinovec et al., 2015). The performance of the Aethalometers during this campaign was investigated by comparing signal from the instruments with TSP and PM$_1$ inlets. The variation not related to the presence of mineral dust was used to determine measurement uncertainty of 11% and 880 nm and 18% at 370 nm (Supplement S3). The influence of the scattering material in the filter matrix, already included in the measurement uncertainty, could be reduced by explicitly taking into account the contribution of the scattering

coefficient to the apparent absorption coefficient. However, this would require the knowledge of the particle size distribution, as the cross-sensitivity to scattering of the filter-based measurement depends on the particle size (Drinovec et al., 2015). The value of the multiple scattering parameter *C* (Weingartner et al., 2003; Drinovec et al., 2015; WMO, 2016) does not add to the final uncertainty because the same value is used for the calibration and the determination of mineral dust concentration, cancelling out in the final calculation.

However, the selection of the parameter *C* influences the calculation of mineral dust absorption coefficient and *MAC*. Similar to parameter *C,* the selected value of *EF* influences determination of absorption coefficients and *MAC*, but not the calculation of mineral dust concentration. It is the variation of *EF*, caused by the changes of the particle size distribution (Figure 5a), which induces about 18% uncertainty. This uncertainty can be reduced by using time-resolved measurements of *EF* or modifying the virtual impactor design to sharpen its response.

The main uncertainty comes from the variability of the chemical composition, mainly from the variability of ratio of Fe/Ca. This ratio is important because *MAC* of mineral dust depends mostly on the iron content, whereas calcium was used as a reference method for determination of mineral dust concentration. The SEM-EDX analysis of single particle chemical composition show large particle-to-particle variation inside the 24 h
filter sample (Supplement S8). As expected the day-to-day variability of chemical composition is much lower as shown by ICP-MS analysis of trace metals (Supplement S7) - we obtained 40% variability of Fe/Ca ratio both for the campaign period as for the year-long dataset.

Desert dust may mix with BC emissions and this is relevant especially at source regions, where concentrations
are large enough for efficient coagulation between dust and BC to occur (Clarke et al., 2004; Rodriguez et al., 2011), with up to a third of carbonaceous particles internally mixed with mineral dust (Hand et al., 2010). The presence of BC on large dust particles will increase the *MAC* of the coarse fraction. The presence of BC on dust means, that for these source regions, larger *MAC* values will be used to convert the optical measurements into dust concentrations. BC present on dust particles contributes negligibly to the mass and the resulting increase
in $PM_{10}$ concentrations is due to dust mass only. The increased *MAC* of these coagulated particles is also the relevant climate parameter, as dust and BC need to be taken into account together when estimating the direct radiative efficiency of such particles. To reduce the uncertainty resulting from different *MAC* values, a mineral dust source location can be determined using back-trajectory analysis and an appropriate *MAC* should be used for each source location.

The combined uncertainty in determination of mineral dust concentration during the Cyprus campaign assuming independent contributions is 44%. The main reason for this uncertainty is the variation of the measured parameters used for the calibration of the method, essentially "assuming the worst-case scenario" of ever-changing aerosolized dust composition and resulting in an overestimation of the uncertainty.
Alternatively, it is possible to derive the uncertainty from the measurement accuracy: to compare daily mineral dust concentrations obtained using VI-PM1 method with the reference values obtained using mass closure. This compares different methods measuring the same sample. Standard deviation of the ratio between predicted and reference mineral dust concentration was 29%. This is a quantification of the scatter of the regression ($R^2$ = 0.92) between the mineral dust concentrations determined using the two methods, as seen in
Figure 7.

**4. Conclusions**

We have demonstrated the potential of the method by showing its applicability at a regional background site in
Agia Marina Xyliiatou (Cyprus), frequently impacted by desert dust. We have shown how to determine the sample *MAC* and use it to quantify with high time resolution the contribution of desert dust to local $PM_{10}$ concentrations.

Although black carbon contribution to the coarse mode is expected to be very small, internal mixture of dust
and black carbon may potentially affect the *MAC* values determined by our methodology. On the other hand, it will not alter the capacity of our methodology to deliver high time resolution $PM_{10}$ concentrations of dust. Instead, our measurement system will be calibrated with such aerosol mixture and a site-specific *MAC* value will be derived, that takes into account this mixing state. Our approach is particularly relevant when using dust optical properties in climate models which need to account for real-world *MAC* values to determine the heating
of the atmosphere due to these aerosols.

The variability of our calibration methodology and therefore, the range of experimentally determined *MAC* values is currently investigated through long-term (multi-year) continuous observations at two regional background sites of the Mediterranean: Agia Marina Xyliatou (Cyprus) and Montseny (Spain). Such data will
offer the unique opportunity to explore the factors controlling dust *MAC* values and in particular the influence

of mineralogy of the different source regions (and especially their hematite and goethite content) and the potential impact of complex mixture of dust with black carbon.

Main conclusions are the following:


- An on-line method (named VI-PM1) for the determination of mineral dust concentration in ambient air based on absorption of coarse particles was developed.
- The VI-PM1 method was calibrated using mass closure performed on 24h filter samples yielding the uncertainty between 29% and 44%, using measurement accuracy and variation of the measured
parameters, respectively.
- The VI-PM1 method allows for easy quantification of mineral dust in environments, where dust absorption is otherwise masked by absorption by black carbon in the fine aerosol fraction.
- During the campaign, we observed a continuous presence of mineral dust with an average of 8 $\mu$g m$^{-3}$ and several intense events with concentrations up to 45 $\mu$g m$^{-3}$.
- An average $MAC_{mineral\_dust,370nm}$ of 0.24 $\pm$ 0.01 m$^2$ g$^{-1}$ and Ångström exponent of 1.41 $\pm$ 0.29 were obtained for mineral dust measured at a background location in Cyprus. This seems to indicate that coarse fraction might be contaminated by black carbon.

*Data availability*. Campaign data can be accessed at the data repository of the Cyprus Institute using the following link: https://mybox.cyi.ac.cy/public.php?service=files&t=624a471bf356165df49cad6cc747b051.

*Competing interests.* Luka Drinovec and Griša Močnik were, at the time of the campaign, but not analysis and writing of the manuscript, also employed by the manufacturer of the Aethalometer AE33 Aerosol d.o.o.
(Slovenia). The methodology was protected with the patent application.

*Acknowledgements.* This work was supported by Slovenian research agency (grant BI-FR/CEA/15-17-004 and programme P1-0099), and Slovenian Ministry of Economic Development and Technology, project DNAAP.

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
