# Peer review of "A new optical-based technique for real-time measurements of mineral dust concentration in PM10 using a virtual impactor"

_Atmospheric Measurement Techniques, 2019_

## Referee Comment (RC1) · Anonymous Referee #1 · 17 Mar 2020

The manuscript presents a novel and interesting technique to quantify the mineral dust concentration starting from calibrated multi-wavelengths absorption measurements. This manuscript deserves to be published in AMT.

Below my comments.

**General comments**

1) The proposed technique assumes that BC particles are not contained in the coarse fraction. Can the authors explain how much is this assumption true? And how the presence of coarse BC affects the mineral dust concentration calculated from the proposed method? There is a comment about this point only at lines 510-514 in the manuscript.

2) The authors should explain better the concept of using a virtual impactor to concentrate coarse particles in the AE33 tape. What it does actually mean? Why the simple formula Abs_TSP minus Abs_PM1 cannot be used to estimate the absorption by dust in the UV? Is the use of the virtual impactor (VI) the way to increase the sensitivity of the technique for measuring the absorption by dust? And from this the need to estimate the enhancement factor (EF)? See another comment on this below.

3) A mean value of 11 for the EF is used here. However, the estimated EF varies quite a lot (Figure 5b). Would it be feasible to measure on-line the EF? How much the uncertainty of EF affects the final result compared to the other sources of error? EF uncertainty affects the MAC uncertainty and consequently the mineral dust concentration estimated from the optical measurements (?).

4) The on-line compensation for filter loading works well (as expected) for PM1 but not for the AE33 connected to the VI. The correction is performed off-line and a mean $k$ is determined and used to correct the data. This introduces an additional uncertainty. Please, comment on this.

5) Figure 6: Is the slight overestimation of the theoretical curve (red line) at 660 nm compared to the experimental one (black line) due to dust absorption in the near-IR? The fit is performed from 880 nm assuming an AAE of 1. Dust should also absorb in the IR, isn't it?

6) Please, expand the Conclusion section if possible.

**Abstract**

Line 30: "......highly time resolved on-line *detection* technique of *dust absorption*...". And also *dust concentration*? Maybe it would be better using the word "*quantification*" rather than "*detection*"?

**Introduction**

Line 56: "Desert dust impacts industrial production to a degree that has been fictionalized (Herbert, 1965)...". Interesting citation, but not very pertinent with the scientific data presented here. What do the authors mean for "industrial production"?

Line 68: "Daily time resolution of the described method has been validated.....". Please, describe briefly the method.

Line 92: "......can be used to identify dust events....". Another advantage of using high-time resolution optical measurements, as the Angstrom Exponent of SSA, is the possibility to detect dust resuspending in the local atmosphere after the end of the episode (when there are no more air masses coming from African deserts). This has big implication for air quality. However, the Angstrom Exponent of SSA cannot be used to quantify the dust concentration whereas the proposed technique does. Maybe a comment on this could be added to the Introduction and Conclusion sections.

Lines 95-120: Here the authors present some previous techniques that can be used to quantify dust concentration in PM using dust aerosol absorption properties. It would be useful to explain what are the main advantages of using the technique proposed in this manuscript compared to the previous techniques. In which terms is the proposed technique innovative? For example, the proposed technique is also affected by problems such as the enhancement due to scattering from dust particles in the filter matrix. A sentence in the Abstract and/or Introduction could be useful.

**Section 2.2.2**

Line 208: ".... the obtained data was compensated using fixed $k$ values as described in ...". Please, define "$k$" here too.

Lines 211-223 (equations 1 and 2): Please, provide more details about the constants presented in this part of the text. For example, how was the new BC MAC at 880 nm calculated? From 3.5/2.14 = 7.77/4.74? What is the constant 1.57? Which type of filter tapes were used during the experiment?

Line 226: The uncertainty of BC was estimated from the plots in Figure S7 as the standard deviation of the ratio BC_tsp/BC_pm1. From equation 2, this ratio is equal to bATN_tsp/bATN_pm1 for each wavelength. Consequently, this ratio should change depending on the relative proportion of dust and BC in the sampled air (especially in the UV). The higher the mineral dust concentration, the higher should be the ratio (at least in the UV). Isn't it? However, no correlation in observed in Figure S7a. Could the authors explain better this point? If I well understand, the lack of correlation in Figure S7a is because the absorption is dominated by BC in TSP too (not only in PM1) due to the much higher MAC of BC compared to dust particles. From this, the need of using a VI to concentrate the dust particles (whereas the BC particles are collected with a CE of unity in the VI) thus allowing measuring the absorption properties of dust. This point is commented at lines 480-485. Maybe it would be better commenting this before, e.g. in the section 2.2.1 or section 3.1.
However, based on the numbers provided in the manuscript, the dust absorption in TSP during the campaign is not negligible compared to the BC absorption: Thus, at 370 nm:
<mineral dust>*MAC_dust = Abs_dust (8.1*0.24=**1.94**)

\ *MAC_BC = Abs_BC (0.39*11.2=**4.4**). However, in Figure S7a a mineral dust concentration value of 15 mg/m3 will lead to a dust absorption at 370 nm of **3.6** Mm-1. Please, comment about this point.

Line 336: "As shown in Figure 3 the absorption in TSP is closely related to the one of PM1". Please, see my previous comment.

**Section 3.4**

Figure 8b: The scatterplot was done using the mineral dust content in $PM_{2.5-10}$. How is the scatterplot if $PM_{10}$ is used instead of $PM_{2.5-10}$? I'm asking in case only $PM_{10}$ is analysed.

Line 471: The MAC from the fit is provided with a very low uncertainty (0.01). isn't too low? It should be higher given that there are important sources of error (BC, EF, $k$, ....). Maybe the fit should be performed with X-Y error bars.

Last (but not least) comment: Would it be possible to test this technique by simulating the mineral dust concentration (and comparing with the filters) using AE33 data collected during other dust events than those presented here? The data presented here are from April-May 2016. Do the authors have other (e.g. more recent) data (AE33 PM1 and VI) available to test the technique using the MAC and EF estimated in this manuscript?

---

## Referee Comment (RC2) · Anonymous Referee #3 · 24 Mar 2020

The paper presents a new technique relying on combining a virtual impactor with an eathalometer, and supports it with a substantial body of laboratory and field data, including both instrument characterization and intercomparisons. The technique under development has the potential to bring significant improvement to dust monitoring and characterization, especially in the context of mixed aerosol types.

The text is well written and informative on the whole, but some stylistic detail cshould be improved, for example inconsistent use of different grammatical tenses (present, present perfect and past), making it difficult to differentiate between the authors' and previous work.

[Figure]

Some detail is not clear, potentially even leading to misunderstanding. For example, the introductory section is somewhat intidy, lacking a logical progression, and ought to be improved.

The statement in lines 204-209 is unclear: is the nonlinearity due to filter saturation? And what are the "k values" - do they compensate for the nonlinearity? Are they constant over time, wavelength etc?

In the Supplement, tests of the virtual impactor (VI) are carried out with PSL, which has specific gravity close to that of water, less that half of that of typical mineral dust. Consequently, the geometric diameter of the latter is substantially smaller than the aerodynamic diameter (relevant in the context of the VI). This aspect is not highlighted and it is not always clear which diameter is discussed. Consequently, a reader using the reported (aerodynamic?) diameters could be misled into applying them to geometric dust sizes.

A constant value of the enhancement factor (EF) seems to be used. Yet EF is variable, as the authors' own data shows, and it will depend on particle aerodynamic size, hence the composition of the sample at any given time. This may be a major shortcoming, affecting the accuracy of the technique. Is it feasible to improve accuracy by using this dependence, perhaps taking advantage of on-line data? While this may not be possible with the current setup alone, the authors should comment on it and suggest potential solutions.

My general concern is about an unspoken shortcoming of the technique: it would fail if the dust and black carbon was internally mixed. As a warning to potential users, this should be highlighted, and the "climatology" of internal as opposed to external mixing described from known historical data.

Another absence is lack in the discussion (or introduction) of comparison of advantages and shortcomings with other methods, such as optical particle counting and aerodynamic sizing.

Typos and corrections: Line 76: "allow for hourly" is written but "allow hourly" is meant. Line 324: "Single" should be "single".

---

## Referee Comment (RC3) · Anonymous Referee #2 · 28 Mar 2020

This work presents a novel on-line detection technique of dust absorption (named VI-PM1) by comparing a coupled high flow virtual impactor sampler with an Aethalometer (model AE33) with the absorption of the submicron aerosol fraction measured with the same absorption photometer. This method was applied for detecting desert dust and was tested in the field for a period of two months at a regional background site in the Eastern Mediterranean. Such new techniques are most valuable to the field and VI-PM1 is expected to provide valuable information about dust particles and their properties. The authors however need to emphasize more the limitations of the method especially under conditions that the mineral dust is contaminated with black carbon.

[Figure]

Specific Comments

L 195: The effect of water uptake in the sampling line, as could occur in Cyprus in spring time due to high ambient temperatures and RH, would require a drying step prior to the aethalometers which is however not described here. On the other hand, the APS and the nephelometer were connected to a nafion providing measurements in dry conditions. How were data handled since different conditions were applied? What limitations may be introduced due to water uptake by the particles?

L 341: In Fig. 3 there are some periods when PM1 measurements seem to be higher than TSP (e.g. 19 April). Can this be attributed to the unit to unit variability?

L. 350: Since the laboratory tests for the enhancement factor were originally performed with flows 75 and 1.5 lpm and 95 and 5 lpm, why did the flows were finally chosen to be 100 and 2 lpm?

L. 357: It would be better the axis to be in $\mu$m rather than nm.

L 410: To my opinion babs, mineral dust is not appropriate to describe the right term of this equation. This would require that there is no BC in the coarse mode, it would fail to describe the possibility of internally mixed BC and dust particles. Once internally mixed, the particles would have different optical properties than those of pure dust (e.g. Scarnato et al., 2015). In the Eastern Mediterranean such a mixture is possible. I recommend to change the left term to bcoarse or similar. This applies to Equation 7 and the subscript of MAC as well. Overall, a short discussion should be dedicated to this issue, expanding the sentence in Line 510 and on.

References

Scarnato, B. V., China, S., Nielsen, K., and Mazzoleni, C.: Perturbations of the optical properties of mineral dust particles by mixing with black carbon: a numerical simulation study, Atmos. Chem. Phys., 15, 6913–6928, https://doi.org/10.5194/acp-15-6913-2015, 2015.

---

## Author Comment (AC1) · 4 Jun 2020

amt-2019-506 - Answer to referee #1 (RC1)

Author's response: We thank the referee for her/his comments which have enabled us to improve the manuscript.

The manuscript presents a novel and interesting technique to quantify the mineral dust concentration starting from calibrated multi-wavelengths absorption measurements. This manuscript deserves to be published in AMT. Below my comments.

General comments

[Figure]

1) The proposed technique assumes that BC particles are not contained in the coarse fraction. Can the authors explain how much is this assumption true? And how the presence of coarse BC affects the mineral dust concentration calculated from the proposed method? There is a comment about this point only at lines 510-514 in the manuscript.

Author's response: As highlighted by the reviewer, possible influence of BC on absorbing properties of coarse particles is an important point to be addressed in the paper. At first, BC concentrations in the coarse mode are expected to be very low, typically ranging 0.02-0.03 $\mu$g/m3, which is 10 times lower compared to fine BC at remote marine locations of the Western/Eastern Mediterranean (Sciare et al., 2003; Mallet et al., 2016). Also, the lack of relationship between the babs,370nm (PM1) and babs,370nm (VI) (Figure 3) together with the strong relationship between babs,370nm (VI) and mineral dust (Figure 7) further supports the idea that dust - rather than BC - is controlling absorption in the coarse mode. Nevertheless, the proposed technique does not make any assumptions whether a fraction of BC is contained or not in the coarse fraction. In other words, the methodology does not exclude possible BC coagulating onto dust aerosols. The EF and MAC determined during the calibration campaign are site-specific (i.e. they are dependent of the dust source region). The MAC for Middle East, for example, would reflect the contamination of dust with intense black carbon emissions that is typical of the region. Variability of MAC results in the increased uncertainty of PMdust as discussed in chapter 3.7. To decrease uncertainty in the determination of mineral dust concentration, different MAC values should be used for different dust source regions.

Changes to the manuscript: Section 3.7, Line 589: "Desert dust may mix with BC emissions and this is relevant especially at source regions, where concentrations are large enough for efficient coagulation between dust and BC to occur (Clarke et al., 2004; Rodriguez et al., 2011), with up to a third of carbonaceous particles internally mixed with mineral dust (Hand et al., 2010). The presence of BC on large dust particles will increase the MAC of the coarse fraction. The presence of BC on dust means, that for

these source regions, larger MAC values will be used to convert the optical measurements into dust concentrations. BC present on dust particles contributes negligibly to the mass and the resulting increase in PM10 concentrations is due to dust mass only. The increased MAC of these coagulated particles is also the relevant climate parameter, as dust and BC need to be taken into account together when estimating the direct radiative efficiency of such particles. To reduce the uncertainty resulting from different MAC values, a mineral dust source location can be determined using back-trajectory analysis and an appropriate MAC should be used for each source location."

2) The authors should explain better the concept of using a virtual impactor to concentrate coarse particles in the AE33 tape. What it does actually mean? Why the simple formula Abs_TSP minus Abs_PM1 cannot be used to estimate the absorption by dust in the UV? Is the use of the virtual impactor (VI) the way to increase the sensitivity of the technique for measuring the absorption by dust? And from this the need to estimate the enhancement factor (EF)? See another comment on this below.

Author's response: An explanation of the concept is added at the beginning of the section 3.

Changes to the manuscript: Section 3, Line 313: "The absorption of the fine aerosol fraction, dominated by BC, is usually much larger compared to the absorption in the coarse fraction, which contains mineral dust. In other words, Abs(TSP) is expected to be close to Abs(PM1). Subtracting these two signals close in absolute values (but with large uncertainties), would result in a close-to-zero number associated with a large measurement error. The proposed VI method takes advantage of the concentration of coarse particles using a virtual impactor to enhance the coarse fraction in the sample, and subtracts the absorption of the fine fraction (as the VI sample contains the same amount of fine fraction as an ambient sample). To calculate the absorption of the coarse fraction, the concentration efficiency of the virtual impactor must be taken into account. "

[Figure]

3) A mean value of 11 for the EF is used here. However, the estimated EF varies quite a lot (Figure 5b). Would it be feasible to measure on-line the EF? How much the uncertainty of EF affects the final result compared to the other sources of error? EF uncertainty affects the MAC uncertainty and consequently the mineral dust concentration estimated from the optical measurements (?).

Author's response: We determined a mean value of EF being 11 with standard deviation of 2. The uncertainty of the calculated mineral dust concentration caused by using a constant value of EF is evaluated in Chapter 3.7: it is 18%. This value is comparable to the absorption measurement uncertainty of the AE33, but smaller than the 40% variability of the mineral dust chemical composition traced by the Fe/Ca ratio. The uncertainty in the determination of MAC depends on the standard error of EF, which is very small (0.023). On-line measurements of EF are possible, but would require additional instrumentation, for example an APS, and additional VI system not necessarily present at the sites of interest.

Changes to the manuscript: Section 3.7, Line 378: "This uncertainty can be reduced by using time-resolved measurements of EF or modifying the virtual impactor design to sharpen its response."

4) The on-line compensation for filter loading works well (as expected) for PM1 but not for the AE33 connected to the VI. The correction is performed off-line and a mean k is determined and used to correct the data. This introduces an additional uncertainty. Please, comment on this.

Author's response: Using constant parameter k value introduces additional uncertainty, but it is very difficult to assess it quantitatively. We attempt to quantify it as a part of the 18% uncertainty assumed for the AE33 measurements at 370 nm (Chapter 3.7).

5) Figure 6: Is the slight overestimation of the theoretical curve (red line) at 660 nm compared to the experimental one (black line) due to dust absorption in the near-IR? The fit is performed from 880 nm assuming an AAE of 1. Dust should also absorb in

the IR, isn't it?

Author's response: Certain species which compose mineral dust show strong absorption in the UV-VIS part of the spectrum, the absorption in infrared is small (Utry et al., 2015). It is possible that the coarse fraction contains some black carbon contamination. If mineral dust would be externally mixed with fine black carbon, we would expect AAE of 1 in the red-IR spectral region. For mineral dust contaminated with black carbon the absorption depends a lot on mineral particle size (Scarnato et al., 2015); the spectral dependence of black carbon stuck to the surface of the mineral dust particles should be evaluated numerically to determine AAE. It is also possible that the scattering artefact of the filter measurement is responsible for the apparent absorption in the infrared. In filter photometers scattering adds to the measured attenuation so that about 1-2% of the scattering coefficient is seen as apparent absorption (Drinovec et al, 2015).

6) Please, expand the Conclusion section if possible.

Author's response: Section 4 was expanded.

Changes to the manuscript: Section 4, Line 614: "We have demonstrated the potential of the method by showing its applicability at a regional background site in Agia Marina Xyliatou (Cyprus), frequently impacted by desert dust. We have shown how to determine the sample MAC and use it to quantify with high time resolution the contribution of desert dust to local PM10 concentrations.

Although black carbon contribution to the coarse mode is expected to be very small, mixture of dust and black carbon may potentially affect the MAC values determined by our methodology. On the other hand, it will not alter the capacity of our methodology to deliver high time resolution PM10 concentrations of dust. Instead, our measurement system will be calibrated with such aerosol mixture and a site-specific MAC value will be derived, that takes into account this mixing state. Our approach is particularly relevant when using dust optical properties in climate models which need to account for real-world MAC values to determine the heating of the atmosphere due to these aerosols.

The variability of our calibration methodology and therefore, the range of experimentally determined MAC values is currently investigated through long-term (multi-year) continuous observations at two regional background sites of the Mediterranean: Agia Marina Xyliatou (Cyprus) and Montseny (Spain). Such data will offer the unique opportunity to explore the factors controlling dust MAC values and in particular the influence of mineralogy of the different source regions (and especially their hematite and goethite content) and the potential impact of complex mixture of dust with black carbon. "

Abstract

Line 30: "......highly time resolved on-line detection technique of dust absorption...". And also dust concentration? Maybe it would be better using the word "quantification" rather than "detection"?

Author's response: We agree.

Changes to the manuscript: Abstract, Line 28: "We build on previous work using filter photometers and present here for the first time a highly time resolved on-line technique for quantification of mineral dust concentration by coupling a high flow virtual impactor (VI) sampler that concentrates coarse particles with an aerosol absorption photometer (Aethalometer, model AE33)."

Introduction

Line 56: "Desert dust impacts industrial production to a degree that has been fictionalized (Herbert, 1965)...". Interesting citation, but not very pertinent with the scientific data presented here. What do the authors mean for "industrial production"?

Author's response: We have expanded the Introduction to highlight the economic effects exerted by dust. We have added the new references.

Changes to the manuscript: Chapter 1, Line 54. "Dust deposits on snow and ice increase the ion content in snow and snow water (Greilinger et al., 2018) and they exert a warming influence after deposition (Di Mauro et al., 2015). Desert dust impacts our

health and economy. Saharan dust events have been shown to increase morbidity and have negative influence on health mainly through respiratory and cardiovascular effects (Middleton et al., 2008; Perez et al., 2012). The health effects of mineral dust are being considered in the context of regulation (WHO, 2018). Dust soiling of photovoltaics is a significant factor in energy production and decreases their output by up to several percent (Mani and Pillai, 2010). Desert dust is a hazard for air and road transport, can cause electric fields detrimental for communication, and impacts water quality and plants, when deposited, resulting in great economic cost (Middleton, 2017), leading to the fictionalization due to its importance (Herbert, 1965)."

Changes to the manuscript: References, line 777. Mani, M. and Pillai, R.: Impact of dust on solar photovoltaic (PV) performance: Research status, challenges and recommendations, Renew. Sust. Energ. Rev., 14, 3124-3131, https://doi.org/10.1016/j.rser.2010.07.065, 2010.

Changes to the manuscript: References, line 784. Middleton, N. J.: Desert dust hazards: A global review, Aeol. Res., 24, 56-63, https://doi.org/10.1016/j.aeolia.2016.12.001, 2017.

Line 68: "Daily time resolution of the described method has been validated…..". Please, describe briefly the method.

Changes to the manuscript: Section 1, Line 71. "Daily time resolution of the described method has been validated with the chemical composition and positive matrix factorization (PMF): the PM10 concentration above the daily regional background monthly 40th percentile has been shown to correlate well with aluminum (as a tracer of mineral dust), and the mineral dust factor from a PMF analysis (Viana et al., 2010).

Line 92: "……can be used to identify dust events….". Another advantage of using high-time resolution optical measurements, as the Angstrom Exponent of SSA, is the possibility to detect dust resuspending in the local atmosphere after the end of the episode (when there are no more air masses coming from African deserts). This has

big implication for air quality. However, the Angstrom Exponent of SSA cannot be used to quantify the dust concentration whereas the proposed technique does. Maybe a comment on this could be added to the Introduction and Conclusion sections.

Author's response: We have added additional text to the Introduction.

Changes to the manuscript: Section 1, Line 96: "These measurements with high time resolution have shown that the optical properties can be used to identify dust events. Additionally, combining the back-trajectory analysis and the SSA wavelength dependence, one can possibly detect local resuspension of dust, which impacts local air quality. However, these methods cannot determine the contribution of desert dust to PM10 concentrations in a quantitative manner.

Lines 95-120: Here the authors present some previous techniques that can be used to quantify dust concentration in PM using dust aerosol absorption properties. It would be useful to explain what are the main advantages of using the technique proposed in this manuscript compared to the previous techniques. In which terms is the proposed technique innovative? For example, the proposed technique is also affected by problems such as the enhancement due to scattering from dust particles in the filter matrix. A sentence in the Abstract and/or Introduction could be useful.

Author's response: We extended a sentence in the Abstract and added additional explanation to the Introduction and uncertainty analysis (chapter 3.7).

Changes to the manuscript: Abstract, Line 28: "We build on previous work using filter photometers and present here for the first time a highly time resolved on-line technique for quantification of mineral dust concentration by coupling a high flow virtual impactor (VI) sampler that concentrates coarse particles with an aerosol absorption photometer (Aethalometer, model AE33)."

Changes to the manuscript: Section1, Line 129: "Previous work has used two-component models to infer dust concentrations sampling ambient air on a filter in filter

absorption photometers. However, the determination of the optical absorption of pure mineral dust - when mixed with black carbon - is more difficult. . ."

Changes to the manuscript: Section 3.7, Line 568: "The influence of the scattering material in the filter matrix, already included in the measurement uncertainty, could be reduced by explicitly taking into account the contribution of the scattering coefficient to the apparent absorption coefficient. However, this would require the knowledge of the particle size distribution, as the cross-sensitivity to scattering of the filter-based measurement depends on the particle size (Drinovec et al., 2015)."

Section 2.2.2

Line 208: ". . .. the obtained data was compensated using fixed k values as described in . . .". Please, define "k" here too.

Changes to the manuscript: Section 2.2.2, Line 213: "Given that the on-line filter loading compensation was not working efficiently for the AE33 coupled with the virtual impactor (see section 3.3, below), the obtained data was compensated using fixed filter loading compensation parameter k values as described in the Supplement S2. "

Lines 211-223 (equations 1 and 2): Please, provide more details about the constants presented in this part of the text. For example, how was the new BC MAC at 880 nm calculated? From 3.5/2.14 = 7.77/4.74? What is the constant 1.57? Which type of filter tapes were used during the experiment?

Author's response: The AE33 measurement during the campaign were conducted using a filter described in Drinovec et al. (2015), as noted in the manuscript. The multiple-scattering parameter C in Drinovec et al. (2015) determined the AE33 filter C values relative to the value of the quartz filter, used in older AE31 instruments. This AE31 value was assumed to be 2.14 (Weingartner et al., 2003), but it was later recommended to use an AE31 value of 3.5 (WMO, 2016). We therefore renormalized the AE33 filter C value:

Cnew = Cold*3.5/2.14 = 2.57

$\sigma$air,new = $\sigma$air,old*Cold/Cnew = 7.77 m2g-1 * 1.57/2.57 = 4.74 m2g-1.

Changes to the manuscript: Section 2.2.2, Line 223: ÂżThe calculation of the absorption coefficient was updated from the Drinovec et al. (2015) following the WMO guideline (WMO, 2016): we updated the value of the filter multiple-scattering parameter C. The multiple-scattering parameter C in Drinovec et al. (2015) determined the AE33 filter C values relative to the value of the quartz filter, used in older AE31 instruments. This AE31 value was assumed to be 2.14 (Weingartner et al., 2003), but it was later recommended to use an AE31 value of 3.5 (WMO, 2016). The parameter C = 1.57 used for the AE33 filter (Drinovec et al., 2015) was renormalized using the same factor resulting in a new value C = 2.57. The mass absorption cross-section $\sigma$air for black carbon was adjusted in the inverse manner to obtain the same BC. The new mass absorption cross section for black carbon $\sigma$air at 880 nm is now 4.74 m2 g-1 instead of 7.77 m2 g-1:Âń

Line 226: The uncertainty of BC was estimated from the plots in Figure S7 as the standard deviation of the ratio BC_tsp/BC_pm1. From equation 2, this ratio is equal to bATN_tsp/bATN_pm1 for each wavelength. Consequently, this ratio should change depending on the relative proportion of dust and BC in the sampled air (especially in the UV). The higher the mineral dust concentration, the higher should be the ratio (at least in the UV). Isn't it? However, no correlation in observed in Figure S7a. Could the authors explain better this point?

Author's response: BC1TSP signal should be higher than BC1PM1 due to the absorption of mineral dust present in the coarse fraction. This is reflected in the slight positive slope of the BC1TSP/BC1PM1 seen in Figure S7a. Because of the Aethalometer measurement uncertainty the fitting parameter error is very large: Slope = 0.0054 +/- 0.0087 $\mu$g-1 m3. We qualitatively explain this as a very small contribution of dust to absorption in TSP relative to BC, measured with a relatively large uncertainty. This is
also the reason why virtual impactor inlet is necessary to concentrate the dust in order to measure its absorption. The reviewer notes this just below.

If I well understand, the lack of correlation in Figure S7a is because the absorption is dominated by BC in TSP too (not only in PM1) due to the much higher MAC of BC compared to dust particles. From this, the need of using a VI to concentrate the dust particles (whereas the BC particles are collected with a CE of unity in the VI) thus allowing measuring the absorption properties of dust. This point is commented at lines 480-485. Maybe it would be better commenting this before, e.g. in the section 2.2.1 or section 3.1.

Author's response: The need to concentrate coarse particles is highlighted in Section1. Line 129. An additional explanation was added at the beginning of section 3.

Changes to the manuscript: Section 3, Line 313: "The absorption of the fine aerosol fraction, dominated by BC, is usually much larger compared to the absorption in the coarse fraction, which contains mineral dust. In other words, $b_{abs,TSP}$ is expected to be close to $b_{abs,PM1}$. Subtracting these two signals close in absolute values (but with large uncertainties), would result in a close-to-zero number associated with a large measurement error. The proposed VI method takes advantage of the concentration of coarse particles using a virtual impactor to enhance the coarse fraction in the sample, and subtracts the absorption of the fine fraction (as the VI sample contains the same amount of fine fraction as an ambient sample). To calculate the absorption of the coarse fraction, the concentration efficiency of the virtual impactor must be taken into account."

However, based on the numbers provided in the manuscript, the dust absorption in TSP during the campaign is not negligible compared to the BC absorption: Thus, at 370 nm:

<mineral dust>*MAC_dust = Abs_dust (8.1*0.24=1.94)

\ *MAC_BC = Abs_BC (0.39*11.2=4.4).

However, in Figure S7a a mineral dust concentration value of 15 mg/m3 will lead to a dust absorption at 370 nm of 3.6 Mm-1. Please, comment about this point.

Author's response: For simplicity we will discuss the topic using absorption coefficients in Table 2 which provides campaign averages for babs,370nm of 6.1 Mm-1 for TSP compared to 5.6 Mm-1 for PM1 which show 9% difference. This difference is much smaller than the uncertainty of 36%. For average PM1 absorption of 5.6 Mm-1 the difference should became statistically significant at mineral dust concentrations above 8.4 $\mu$g/m3.

Changes to the manuscript: Section 3.3, Line 465: "The average difference between absorption for TSP and PM1 inlets is lower than expected from the (VI-PM1)/2 absorption value. This is a consequence of high measurement uncertainty of up to 18% during the campaign (as estimated comparing Aethalometers with different inlets, Supplement S3). Mineral dust absorption calculated from the difference between babs,370nm for TSP and PM1 of 0.6 Mm-1 has an uncertainty 2.1 Mm-1."

Line 336: "As shown in Figure 3 the absorption in TSP is closely related to the one of PM1". Please, see my previous comment.

Author's response: The analysis in Supplement S3 shows that there is no significant difference between the absorptions measured by two Aethalometers with TSP and PM1 inlets.

Changes to the manuscript: Section 3.1, Line 356: "The differences are inside the measurement uncertainty of the Aethalometers (Supplement S3)."

Section 3.4

Figure 8b: The scatterplot was done using the mineral dust content in PM2.5-10. How is the scatterplot if PM10 is used instead of PM2.5-10? I'm asking in case only PM10 is analysed.

Author's response: For calculation of MAC we have used mineral dust contained in PM10-2.5 fraction. The resulting mineral dust concentration (equation 7) is the mineral dust concentration in the coarse fraction. To calculate mineral dust in PM10, a correction factor (1.06) should be used as mentioned in the first paragraph of section 3.7. If mineral dust in PM10 is used for the calibration then the resulting mineral dust concentration (equation 7) reports the mineral dust concentration in PM10.

Line 471: The MAC from the fit is provided with a very low uncertainty (0.01). isn't too low? It should be higher given that there are important sources of error (BC, EF, k, ....). Maybe the fit should be performed with X-Y error bars.

Author's response: The MAC is reported with the fitting parameter standard error. The uncertainty of MAC is discussed in detail in chapter 3.7. Because of the variability of Ca and Fe content in mineral dust the uncertainty of MAC is about 40%.

Last (but not least) comment: Would it be possible to test this technique by simulating the mineral dust concentration (and comparing with the filters) using AE33 data collected during other dust events than those presented here? The data presented here are from April-May 2016. Do the authors have other (e.g. more recent) data (AE33 PM1 and VI) available to test the technique using the MAC and EF estimated in this manuscript?

Author's response: The method is currently being tested in Cyprus (Agia Marina station) and Northern Spain (Montseny station). These datasets will allow for determination of MAC and EF for different source locations. Using source-area specific parameters will lower the uncertainty in the determination of mineral dust concentrations. The results will be published in several separate manuscripts in preparation. The Conclusions section was updated.

Changes to the manuscript: Section 4, Line 627: "The variability of our calibration methodology and therefore, the range of experimentally determined MAC values is currently investigated through long-term (multi-year) continuous observations at two

regional background sites of the Mediterranean: Agia Marina Xyliatou (Cyprus) and Montseny (Spain). Such data will offer the unique opportunity to explore the factors controlling dust MAC values and in particular the influence of mineralogy of the different source regions (and especially their hematite and goethite content) and the potential impact of complex mixture of dust with black carbon."

References

Mani, M. and Pillai, R.: Impact of dust on solar photovoltaic (PV) performance: Research status, challenges and recommendations, Renew. Sust. Energ. Rev., 14, 3124-3131, https://doi.org/10.1016/j.rser.2010.07.065, 2010. Middleton, N. J.: Desert dust hazards: A global review, Aeol. Res., 24, 56-63, https://doi.org/10.1016/j.aeolia.2016.12.001, 2017.

Weingartner, E., Saathoff, H., Schnaiter, M., Streit, N., Bitnar, B., and Baltensperger, U.: Absorption of light by soot particles: determination of the absorption coefficient by means of aethalometers, J. Aerosol Sci., 34, 1445–1463, doi:10.1016/S0021-8502(03)00359-8, 2003. WMO, Report No. 227: WMO/GAW Aerosol Measurement Procedures, Guidelines and Recommendations, 2nd Edition, 2016, Geneva, WMO.

---

## Author Comment (AC2) · 4 Jun 2020

amt-2019-506 - Answer to referee #3 (RC2)

Author's response: We thank the referee for her/his comments which have enabled us to improve the manuscript.

The paper presents a new technique relying on combining a virtual impactor with an eathalometer, and supports it with a substantial body of laboratory and field data, including both instrument characterization and intercomparisons. The technique under development has the potential to bring significant improvement to dust monitoring and

characterization, especially in the context of mixed aerosol types. The text is well written and informative on the whole, but some stylistic detail should be improved, for example inconsistent use of different grammatical tenses (present, present perfect and past), making it difficult to differentiate between the authors' and previous work.

Some detail is not clear, potentially even leading to misunderstanding. For example, the introductory section is somewhat intidy, lacking a logical progression, and ought to be improved.

Author's response: We have taken all reviewer's comments in account and added to the Introduction to make it clearer.

The statement in lines 204-209 is unclear: is the nonlinearity due to filter saturation? And what are the "k values" - do they compensate for the nonlinearity? Are they constant over time, wavelength etc?

Author's response: The non-linearity caused by the filter loading effect is explained in Drinovec et al. (2015) article. The topic is quite extensive, for that reason it is kept out of this article.

Changes to the manuscript: Section 2.2.2, Line 213: "Given that the on-line filter loading compensation was not working efficiently for the AE33 coupled with the virtual impactor (see section 3.3, below), the obtained data was compensated using fixed filter loading compensation parameter k values as described in the Supplement S2. "

In the Supplement, tests of the virtual impactor (VI) are carried out with PSL, which has specific gravity close to that of water, less that half of that of typical mineral dust. Consequently, the geometric diameter of the latter is substantially smaller than the aerodynamic diameter (relevant in the context of the VI). This aspect is not highlighted and it is not always clear which diameter is discussed. Consequently, a reader using the reported (aerodynamic?) diameters could be misled into applying them to geometric dust sizes.

Author's response: Aerodynamic diameter is used throughout the article. For clarity the adjective "aerodynamic" was added in several places. The fact, that virtual impactor characteristic depend on aerodynamic particle size was emphasized at the beginning of the Chapter 3.2.

Changes to the manuscript: Section 3.2, Line 369: "The concentration efficiency of the virtual impactor depends on the aerodynamic particle diameter. For that reason, an Aerodynamic Particle Sizer (APS, model 3321) was used both for analysis of aerodynamic particle size distribution during the campaign and virtual impactor characterization (Supplement S1)."

A constant value of the enhancement factor (EF) seems to be used. Yet EF is variable, as the authors' own data shows, and it will depend on particle aerodynamic size, hence the composition of the sample at any given time. This may be a major shortcoming, affecting the accuracy of the technique. Is it feasible to improve accuracy by using this dependence, perhaps taking advantage of on-line data? While this may not be possible with the current setup alone, the authors should comment on it and suggest potential solutions.

Author's response: The uncertainty introduced by using a constant value of EF for calculation of mineral dust concentration is 18% as noted in Chapter 3.7, page 14, line 578. Using APS for real-time determination of EF would reduce that uncertainty.

Changes to the manuscript: Section 3.7, Line 578: "This uncertainty can be reduced by using time-resolved measurements of EF or modifying the virtual impactor design to sharpen its response."

My general concern is about an unspoken shortcoming of the technique: it would fail if the dust and black carbon was internally mixed. As a warning to potential users, this should be highlighted, and the "climatology" of internal as opposed to external mixing described from known historical data. Author's response: The proposed method is calibrated during the intensive campaign when MAC and EF are determined. Both

parameters depend on the source region: EF is affected by the changes in size distribution which with time moves toward the smaller particles. MAC can be affected by the mineral dust composition, for example iron content and ionic state and also by black carbon content. The coagulation of mineral dust and black carbon is most intense at the source location because the process depends a lot on the aerosol concentration (Rodriguez et al., 2011). During transport the mineral dust and black carbon are diluted slowing the coagulation process. In both cases MAC of the mineral dust depends on the source region. The uncertainty of the method can be greatly reduced if the source region is determined from back-trajectory analysis and a specific MAC value is applied. This subject is being studied and will be reported in a separate article.

Changes to the manuscript: Section 3.7, Line 589: "Desert dust may mix with BC emissions and this is relevant especially at source regions, where concentrations are large enough for efficient coagulation between dust and BC to occur (Clarke et al., 2004; Rodriguez et al., 2011), with up to a third of carbonaceous particles internally mixed with mineral dust (Hand et al., 2010). The presence of BC on large dust particles will increase the MAC of the coarse fraction. The presence of BC on dust means, that for these source regions, larger MAC values will be used to convert the optical measurements into dust concentrations. BC present on dust particles contributes negligibly to the mass and the resulting increase in PM10 concentrations is due to dust mass only. The increased MAC of these coagulated particles is also the relevant climate parameter, as dust and BC need to be taken into account together when estimating the direct radiative efficiency of such particles. To reduce the uncertainty resulting from different MAC values, a mineral dust source location can be determined using back-trajectory analysis and an appropriate MAC should be used for each source location."

Another absence is lack in the discussion (or introduction) of comparison of advantages and shortcomings with other methods, such as optical particle counting and aerodynamic sizing.

Author's response: Particle counting and sizing have not been included in discussion

because these methods do not discriminate between mineral dust and other aerosol. Other qualitative and quantitative methods have been discussed in the Introduction. This discussion has been extended in line with the reviewers' comments.

Typos and corrections: Line 76: "allow for hourly" is written but "allow hourly" is meant. Line 324: "Single" should be "single".

Author's response: The misspellings are corrected. Also the figure numbering was corrected.

References Rodríguez, S., Alastuey, A., Alonso-Pérez, S., Querol, X., Cuevas, E., Abreu-Afonso, J., Viana, M., Pérez, N., Pandolfi, M., and de la Rosa, J.: Transport of desert dust mixed with North African industrial pollutants in the subtropical Saharan Air Layer, Atmos. Chem. Phys., 11, 6663–6685, https://doi.org/10.5194/acp-11-6663-2011, 2011.
* * *

---

## Author Comment (AC3) · 4 Jun 2020

amt-2019-506 - Answer to referee #2 (RC3)

Author's response: We thank the referee for her/his comments which have enabled us to improve the manuscript.

This work presents a novel on-line detection technique of dust absorption (named VI-PM1) by comparing a coupled high flow virtual impactor sampler with an Aethalometer (model AE33) with the absorption of the submicron aerosol fraction measured with the same absorption photometer. This method was applied for detecting desert dust

and was tested in the field for a period of two months at a regional background site in the Eastern Mediterranean. Such new techniques are most valuable to the field and VI-PM1 is expected to provide valuable information about dust particles and their properties. The authors however need to emphasize more the limitations of the method especially under conditions that the mineral dust is contaminated with black carbon.

Author's response: The method uses a single MAC value to calculate mineral dust concentration from the absorption data. MAC for different source locations is affected both by the variation of mineral composition (see chapter 3.7) and also by the possible contamination by black carbon. We treat the mineral dust contaminated by BC as the mineral dust that is measured at the receptor site, including the possible contamination. The uncertainty due to variation of the MAC can be lowered by using source location specific MAC value.

Changes to the manuscript: Section 3.7, Line 594589: "Desert dust may mix with BC emissions and this is relevant especially at source regions, where concentrations are large enough for efficient coagulation between dust and BC to occur (Clarke et al., 2004; Rodriguez et al., 2011), with up to a third of carbonaceous particles internally mixed with mineral dust (Hand et al., 2010). The presence of BC on large dust particles will increase the MAC of the coarse fraction. The presence of BC on dust means, that for these source regions, larger MAC values will be used to convert the optical measurements into dust concentrations. BC present on dust particles contributes negligibly to the mass and the resulting increase in PM10 concentrations is due to dust mass only. The increased MAC of these coagulated particles is also the relevant climate parameter, as dust and BC need to be taken into account together when estimating the direct radiative efficiency of such particles. To reduce the uncertainty resulting from different MAC values, a mineral dust source location can be determined using back-trajectory analysis and an appropriate MAC should be used for each source location. "

Specific Comments

L 195: The effect of water uptake in the sampling line, as could occur in Cyprus in spring time due to high ambient temperatures and RH, would require a drying step prior to the aethalometers which is however not described here. On the other hand, the APS and the nephelometer were connected to a nafion providing measurements in dry conditions. How were data handled since different conditions were applied? What limitations may be introduced due to water uptake by the particles?

Author's response: Certain aerosol species are hygroscopic which cause the increase in particle size in increased RH. The most hygroscopic are salts (sea salt, nitrates and sulphates) and oxygenated organic compounds. Mineral dust and black carbon alone are not hygroscopic and thus (if not coated) not susceptible to changes of optical properties due to humidity changes. Aethalometer samples have not been dried, as stated in the manuscript. The station temperature was kept at 25+/-2 deg. C. Due to the high sample flow the virtual impactor operated at ambient temperature. During the campaign, the average ambient temperature was 20 deg. C with 35% relative humidity. Due to small differences between ambient and station temperatures, we did not apply any correction to the data.

L 341: In Fig. 3 there are some periods when PM1 measurements seem to be higher than TSP (e.g. 19 April). Can this be attributed to the unit to unit variability?

Author's response: Yes, the difference is attributed to the unit-to unit variability, which was quantified in the Supplement S3. It shows that measurement uncertainty at 370 nm is about 18%.

Changes to the manuscript: Section 3.1, Line 360356: ÂżThe differences are inside the measurement uncertainty of the Aethalometers (Supplement S3). Âż

L. 350: Since the laboratory tests for the enhancement factor were originally performed with flows 75 and 1.5 lpm and 95 and 5 lpm, why did the flows were finally chosen to be 100 and 2 lpm?

[Figure]

Author's response: We wanted to obtain higher concentration efficiencies which depend on the ratio of the flows Fin/Fout. With the minimum sample flow through the AE33 of 2 lpm we chose Fin of 100 lpm. We used the calibration curve obtained at 75 lpm to 1.5 lpm which has the same theoretical concentration efficiency of 50.

Changes to the manuscript: Section 3.2, Line 376372: "Using the minimum sample flow of AE33 of 2 lpm and maximum flow of the virtual impactor pump we obtained we obtained the maximum Fin/Fout ratio of 50.Âń

L. 357: It would be better the axis to be in $\mu$m rather than nm.

Author's response: The axis units on Figure 4 was changed to $\mu$m.

L 410: To my opinion babs, mineral dust is not appropriate to describe the right term of this equation. This would require that there is no BC in the coarse mode, it would fail to describe the possibility of internally mixed BC and dust particles. Once internally mixed, the particles would have different optical properties than those of pure dust (e.g. Scarnato et al., 2015). In the Eastern Mediterranean such a mixture is possible. I recommend to change the left term to bcoarse or similar. This applies to Equation 7 and the subscript of MAC as well. Overall, a short discussion should be dedicated to this issue, expanding the sentence in Line 510 and on.

Author's response: The absorption coefficient in Equation 6 represents all absorbing species in the coarse aerosol fraction, thus the caption can be changed from babs, mineral dust to babs, PM10-1. The situation is different for MACmineral dust, 370 nm, which is calculated using reference mineral dust concentration and not coarse mass concentration. In the same way Equation 7 represents calculated mineral dust concentration and not coarse fraction mass concentration.

Changes to the manuscript: Equation 6 is changed to $b_{(abs,PM10-1)}=(b_{(abs,VI)}-b_{(abs,PM1)})/EF$ .

Author's response: As explained above, MAC is a property of the mineral dust sampled

during the calibration campaign. The value is representative for the source region during the campaign. As observed during our measurement campaign, the dust was probably contaminated by black carbon. The uncertainty of the method can be further reduced by applying MAC values specific for each source region. The subject is being studied and will be reported in a separate article.

Changes to the manuscript: Section 3.24, Line 541537:"Simulations show that black carbon stuck to the mineral dust particles can severely change its optical properties, but the effect depends on the particle size (Scarnato et al., 2015). Because of the differences in dust mineral composition and contamination with BC, we expect MAC to be source region specific."

Changes to the manuscript: Section 3.7, Line 594589: "Desert dust may mix with BC emissions and this is relevant especially at source regions, where concentrations are large enough for efficient coagulation between dust and BC to occur (Clarke et al., 2004; Rodriguez et al., 2011), with up to a third of carbonaceous particles internally mixed with mineral dust (Hand et al., 2010). The presence of BC on large dust particles will increase the MAC of the coarse fraction. The presence of BC on dust means, that for these source regions, larger MAC values will be used to convert the optical measurements into dust concentrations. BC present on dust particles contributes negligibly to the mass and the resulting increase in PM10 concentrations is due to dust mass only. The increased MAC of these coagulated particles is also the relevant climate parameter, as dust and BC need to be taken into account together when estimating the direct radiative efficiency of such particles. To reduce the uncertainty resulting from different MAC values, a mineral dust source location can be determined using back-trajectory analysis and an appropriate MAC should be used for each source location."

Changes to the manuscript: References, Line 830832: "Rodríguez, S., Alastuey, A., Alonso-Pérez, S., Querol, X., Cuevas, E., Abreu-Afonso, J., Viana, M., Pérez, N., Pandolfi, M., and de la Rosa, J.: Transport of desert dust mixed with North African industrial pollutants in the subtropical Saharan Air Layer, Atmos. Chem. Phys., 11, 6663–

6685, https://doi.org/10.5194/acp-11-6663-2011, 2011. Changes to the manuscript: References, Line 834836: "Scarnato, B. V., China, S., Nielsen, K., and Mazzoleni, C.: Perturbations of the optical properties of mineral dust particles by mixing with black carbon: a numerical simulation study, Atmos. Chem. Phys., 15, 6913–6928, https://doi.org/10.5194/acp-15-6913- 2015, 2015."

References

Scarnato, B. V., China, S., Nielsen, K., and Mazzoleni, C.: Perturbations of the optical properties of mineral dust particles by mixing with black carbon: a numerical simulation study, Atmos. Chem. Phys., 15, 6913–6928, https://doi.org/10.5194/acp-15-6913-2015, 2015.

References Rodríguez, S., Alastuey, A., Alonso-Pérez, S., Querol, X., Cuevas, E., Abreu-Afonso, J., Viana, M., Pérez, N., Pandolfi, M., and de la Rosa, J.: Transport of desert dust mixed with North African industrial pollutants in the subtropical Saharan Air Layer, Atmos. Chem. Phys., 11, 6663–6685, https://doi.org/10.5194/acp-11-6663-2011, 2011.